# In-Context Impersonation Reveals Large Language Models' Strengths and Biases

Leonard Salewski[1,2]      Stephan Alaniz[1,2]      Isabel Rio-Torto[3,4]*

Eric Schulz[2,5]      Zeynep Akata[1,2]

[1] University of Tübingen      [2] Tübingen AI Center      [3] University of Porto
[4] INESC TEC      [5] Max Planck Institute for Biological Cybernetics

## Abstract

In everyday conversations, humans can take on different roles and adapt their vocabulary to their chosen roles. We explore whether LLMs can take on, that is impersonate, different roles when they generate text in-context. We ask LLMs to assume different personas before solving vision and language tasks. We do this by prefixing the prompt with a persona that is associated either with a social identity or domain expertise. In a multi-armed bandit task, we find that LLMs pretending to be children of different ages recover human-like developmental stages of exploration. In a language-based reasoning task, we find that LLMs impersonating domain experts perform better than LLMs impersonating non-domain experts. Finally, we test whether LLMs' impersonations are complementary to visual information when describing different categories. We find that impersonation can improve performance: an LLM prompted to be a bird expert describes birds better than one prompted to be a car expert. However, impersonation can also uncover LLMs' biases: an LLM prompted to be a man describes cars better than one prompted to be a woman. These findings demonstrate that LLMs are capable of taking on diverse roles and that this in-context impersonation can be used to uncover their strengths and hidden biases. Our code is available at `https://github.com/ExplainableML/in-context-impersonation`.

## 1  Introduction

Large Language Models (LLMs) can not only summarize documents and converse on a large range of topics [1], but they have also shown other emergent abilities [2, 3]. Because of their impressive abilities, LLMs are permeating into many applications [4, 5]. This means that there is a societal need to understand how these models "tick" [6, 7]. Traditionally, LLMs are provided with a context as a textual prompt and are asked to provide answers via text completion, thereby solving a variety of choice-based [8], description-based [9], and reasoning tasks [10]. Yet how in-context learning works is not fully understood. When Min et al. [11] prompted LLMs with random labels, they found that this did not drastically degrade performance, suggesting that the role of in-context demonstrations is to prime the model for a particular task. This is in line with other results suggesting that LLMs internally infer latent variables to make better predictions [12]. It has been suggested that LLMs, and other large models, can change their behavior when asked to respond as a particular persona. When Deshpande et al. [13] asked LLMs to respond as a hateful person, their toxicity score increased. When Wang and colleagues [14] asked LLMs to imagine being expert systematic reviewers, the

---

*Work done during a research visit at the University of Tübingen

37th Conference on Neural Information Processing Systems (NeurIPS 2023).

quality of their literature search queries increased. That LLMs can impersonate specific people is also known; they can, for example, pretend to be Oscar Wilde, Carrie Bradshaw from Sex and the City, or Donald Trump [15]. But how does in-context impersonation affect LLMs' behavior in language-based and other downstream tasks?

In the current work, we let LLMs impersonate, that is taking on different roles, in context. We do this by prefixing the prompt with *"If you were a {persona}"* where *persona* is replaced with the persona that the LLM is asked to impersonate. These personas are associated either with a social identity or a domain of expertise. In a first simulation using a multi-armed bandit task [16], we find that LLMs impersonating children of different ages can recover the developmental stages of human-like exploration strategies. In language-based reasoning tasks, we find that LLMs impersonating domain experts perform better than LLMs impersonating non-domain experts. Finally, we ask LLMs to describe different classes of either birds or cars and then use their descriptions in a downstream, visual classification task. The results of this experiment corroborate our earlier results: LLMs become better as they pretend to be older, and they are also better when they pretend to be domain experts. However, we also see how impersonating LLMs reproduce biases: LLMs impersonating a black person or a male describe cars better, while LLMs impersonating a white person or a female describe birds better. These results expand our understanding of in-context learning in LLMs and open up new research directions investigating role-taking and pretense in LLMs and beyond.

## 2   Related Work

In-context learning refers to an LLM's ability to improve at a given task after being provided with a number of task-relevant demonstrations [1]. This ability sets LLMs apart from traditional models and has led to a totally new paradigm – one which does not require fine-tuning of weights on task-specific data but instead relies entirely on contextual information [17, 10, 18].

This contextual information is normally delivered as textual prompts [19], where a task or scenario is described and a model is asked to solve the task or reason about the scenario by generating the next words of the provided text. Due to its flexibility, prompting has been widely used as a generic method for natural language tasks [20, 21]. Importantly, the resulting in-context learning does not only work after LLMs have seen some examples, i.e. in the few-shot regime [22], but also without any examples, i.e. in the zero-shot regime [23]. LLMs are reasonably proficient at solving arithmetic [24] or reasoning tasks [25] without having been prompted with example solutions but only after being asked to provide an answer to a given problem. LLMs can require careful engineering of the provided prompts, either manually [26] or automatically [27]. Indeed, whole books have been written to provide guidelines on how to best perform prompt engineering [28], especially because engineering prompts can require a great amount of expertise [29].

One method known to influence LLMs behavior is to ask them to respond as a particular person [30, 31], an effect which is also described as role-taking [32]. LLMs can take in the text of one famous author, e.g. Oscar Wilde, and rewrite it in the style of another famous author, e.g. James Joyce [33]. This is not only true for LLMs but for any large model that provides results based on prompts, such as text-to-image models [34–36]. For example, using the artist's name for generative art prompting is known to boost the quality [29] or to substantially affect the style [37–39] of the generated images. To make LLMs respond more truthfully, Lin and colleagues introduced scenarios from the perspective of a fictional persona called "Professor Smith" [40]. Conversely, to make LLMs act maliciously, Wolf et al. [41] prompt LLMs adversarially to overcome alignment techniques. LLMs can also be used to simulate multiple humans which changes how they cooperate in economic games [42].

LLMs can also have their own "personalities" which can be evoked in-context [43]. Although LLMs frequently behave like the average person [44], their personality profiles can be tinkered with [45], e.g. by changing the context to be more or less emotional [46]. This has led researchers to use LLMs to simulate survey responses [47] of subpopulations by conditioning them on socio-demographic descriptions [48] or to ask them to respond in persona when writing about fictitious childhood events [49]. Additionally, non-deterministic tasks such as open-ended questions have also been explored [50].

Semantics derived automatically from language corpora can contain human-like biases [51]. Thus, LLMs do not only reproduce human-like text but also replicate biases present in the training

data [7, 52]. Importantly, these biases can get exacerbated if LLMs are asked to provide answers in persona [46, 13, 53].

LLMs are naturally combined with large vision-language models (VLMs) [54, 55] such as CLIP [56] due to their versatility in a wide range of visual recognition tasks. Menon et al. [57] used GPT-3 [1] to generate a diverse set of short descriptions of a class that improve zero-shot classification when their CLIP scores are combined. Similarly, Yang et al. [58] used GPT-3 descriptions of classes as concept bottlenecks for interpretable image classification. LLMs can also be used as a knowledge base for visual question-answering (VQA) tasks [59].

## 3 In-context Impersonation Methodology

Our methodology is composed of two steps. First, we prompt and query the LLM. Second, we evaluate the resulting text queries in three tasks, i.e. two-armed bandit, reasoning, and visual classification.

### 3.1 Prompting and Querying the Large Language Model with Personas

LLMs are trained to predict the most probable next token $t_k$ given previous tokens $t_1 \ldots t_{k-1}$ by maximizing the likelihood function $p_{\text{LLM}}(t_k | t_1, \ldots, t_{k-1})$. In this work, we use pre-trained LLMs without further finetuning them. Depending on the task, we generate one or more tokens given a task-specific context $c$ that describes the task to the language model and prompts it for an answer. The context includes the instruction to impersonate using the phrase *"If you were a {persona}"* where persona $p$ is replaced by the persona name. Thus, we obtain generated tokens $t$ by sampling from

$$p_{\text{LLM}}(\boldsymbol{t} | \boldsymbol{c}^{(p)}) = \prod_{k=1}^{K} p_{\text{LLM}}(t_k | c_1^{(p)}, \ldots, c_n^{(p)}, t_1, \ldots, t_{k-1}) \qquad (1)$$

We refer to this type of contextualization as *in-context impersonation*.

**Personas Considered.** The first interesting question to look at was if LLMs could impersonate the behavior of differently aged people. For this, we ask the LLM to imagine it is either a 2, 4, 7, 13, or 20-year-old. We also evaluate whether the LLM is able to impersonate different fields of expertise. Depending on the task considered, the expertise profiles differ (more details below). Finally, we evaluate whether LLMs have biases regarding gender and skin color. For this, we asked LLMs to imagine that they were either a man or a woman or a black person or a white person.

**Large Language Models Considered.** In this work, we evaluate two LLMs. For all of our tasks, we used the Vicuna-13B language model [60] which has 13 billion parameters and was trained to follow natural language instructions. Vicuna is a fine-tuned version of the LLAMA language model [61] using ShareGPT [62] conversational data. We use an instruction fine-tuned model because it was optimized to follow user prompts. Its weights are publicly available, allowing us to run the model locally. Vicuna is competitive with proprietary services such as ChatGPT in some domains [63][2]. In addition to Vicuna, we use the OpenAI API of ChatGPT [64] with the `gpt-3.5-turbo` model for the reasoning and vision tasks. For the bandit task, however, running 12k games with 10 trials each is infeasible.

We do not further train the models, nor do we provide sample solutions in-context; thus, all experiments are conducted in a zero-shot fashion. By providing minimal guidance to perform the task, we avoid pre-conditioning the model such that answers can better reflect the internalized language of the LLM instead of relying on few-shot examples. When sampling full sentences, we use a temperature of 0.7; to obtain the answer as a single symbol (token), we set it to 1 unless otherwise stated. These different temperatures were chosen based on the recommended default values of each LLM.

### 3.2 Bandit Task Design

We asked LLMs to imagine being in different personalities while participating in a multi-armed bandit task [65] taken from the psychology literature [66] and already applied to LLMs [8].

An agent gets to interact with a two-armed bandit problem for 10 trials. The mean reward for each arm $a$ is drawn from $p(\theta_a) = \mathcal{N}(0, 10)$ at the beginning of a task, and the reward for each trial

---

[2]`https://chat.lmsys.org/?leaderboard`

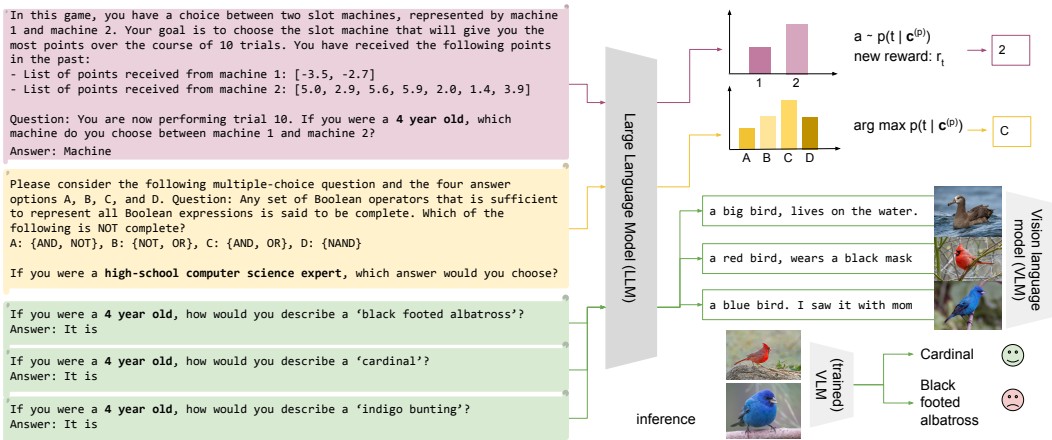

Figure 1: Our three tasks are designed to analyze the effect of *in-context impersonation*. First, we investigate bandit tasks (pink) where the LLM must maximize the reward while impersonating different age groups. Second, we evaluate the effect of domain expert impersonation on natural language reasoning tasks (yellow). Third, we study the usefulness of descriptions generated with impersonation w.r.t. age, expertise, ethnicity, and gender for visual classification (green).

is drawn from $p(r_t|a_t, \theta_{a_t}) = \mathcal{N}(\theta_{a_t}, 1)$. Feedback of past trials is provided via prompt-chaining, i.e. concatenating previous choices and their outcomes to the current prompt submitted to the LLM. We analyze the set of emerging exploration strategies, assuming that an agent uses Bayes' rule to update its beliefs over unobserved parameters. If prior and rewards are normally distributed, then the posterior will be normally distributed and the corresponding updating rule is given by the Kalman filtering equations. Let $p(\theta_a|h_t) = \mathcal{N}(\mu_{a,t}, \sigma_{a,t})$ be the posterior distribution at time-step $t$. Based on the parameters of this posterior distribution, one can define a probit-regression model:

$$p(A_t = 1|\mathbf{w}) = \mathbf{\Phi}\left(\beta_1 V_t + \beta_2 RU_t\right) \tag{2}$$

with $\mathbf{\Phi}$ denoting the cumulative distribution function of a standard normal distribution. Here, $V_t = \mu_{1,t} - \mu_{2,t}$ represents the estimated difference in value and $RU_t = \sigma_{1,t} - \sigma_{2,t}$ the relative uncertainty. One can use Equation 2 to analyze how much an agent engages in exploitation behavior by inspecting $\beta_1$ and how much the agent uses uncertainty to explore in a directed fashion by inspecting $\beta_2$ [16].

For this bandit task, we consider personas of different ages. Specifically, we study ages 2, 4, 7, 13, and 20 to cover key developmental stages of early childhood, childhood, adolescence, and adulthood where the learning progress is most pronounced in humans. The language model is prompted (see Figure 1, the pink path) to only answer "1" or "2" depending on which arm $a$ it would like to choose. The LLM receives rewards and the associated actions from previous trials inside the context in the form of a list.

With $\log d_{a_t} = \log p_{\text{LLM}}(t_1 = a_t|\mathbf{c}^{(p)}, a_1, \ldots, a_{t-1}, r_1, \ldots, r_{t-1})$ being the unnormalized logits from the LLM for the token of arm $a$, for each trial we sample an action $\hat{a} \sim \sigma\left(\{\log d_{a_t}\}_{a_t=1}^{A}\right)$ where we have two arms $A = 2$. We do not apply temperature scaling in this case as we are only sampling a single token and want it to reflect the LLM decision-making as faithfully as possible.

### 3.3 Reasoning Task Design

In our reasoning task, the LLM has to answer a multiple-choice question regarding a given topic from the Multitask Language Understanding (MMLU) dataset [67], commonly used to benchmark LLMs [61]. The MMLU dataset consists of 57 tasks from Science, Technology, Engineering, and Mathematics (STEM), Humanities, Social Sciences, and Other, ranging from elementary, high school, college, and professional levels of complexity. We start by prompting the LLM with the context:

```
Please consider the following multiple-choice question and the four answer
             options A, B, C, and D. Question:  {task}
        If you were a {persona}, which answer would you choose?
```

The *task* is replaced by the question and the 4 possible answers, while the *persona* is replaced by an expert (see Figure 1, the yellow path). We consider three types of experts as personas. The task expert, e.g. for the high school computer science task, is "high school computer science expert". The domain expert is an aggregation of all the remaining experts in the same field as the task expert (but not the task expert himself), e.g. for high school computer science it would be any other STEM expert. The non-domain expert is an aggregation of the task experts from the other domains, e.g. for high school computer science it would be all Humanities, Social Sciences and Other experts.

After feeding the prompt to the LLM, the LLM prediction of the first token following the context is $d = p_{\text{LLM}}(t_1|\boldsymbol{c}^{(p)})$ and the $N$ tokens for the possible answers of the multiple choice question are $o = \{o_i\}_{i=1}^{N}$ which in this case are A, B, C, and D. The predicted option is then given by

$$\hat{o} = \arg\max(\hat{c}_i), \text{ with } \hat{c}_i = d[c_i], i = 1 \ldots N \tag{3}$$

which are the predicted probabilities of the language model. With this approach, we are able to obtain the option with the highest probability according to the LLM and, thus, compare it with the ground truth label to measure the accuracy resulting from different in-context impersonations.

### 3.4 Vision and Language Task Design

Lastly, we want to evaluate the usefulness of descriptions generated by *in-context impersonation* for downstream vision and language tasks. We focus on challenging fine-grained classification tasks, as the generated descriptions need to be domain specific for these tasks to succeed. We ask the LLMs to generate a description of a class, from the perspective of a persona. Our prompt is:

```
If you were a {persona}, how would you answer the following question
         in 45 words?  Q: What is a/an {class_name}?  A: It is
```

To avoid trivial solutions, i.e. the class name being mentioned in the description, we post-process the generated descriptions with a two-step approach: first, we replace class names used in noun phrases with an appropriate pronoun whilst respecting the given numerous. Second, if the class name is still not removed, we re-use the same language model to process the descriptions sentence by sentence. For this, we use 4 in-context examples, that demonstrate how to remove the class name information. The full process is documented in suppl. Section D.1.

**Vision-Language Models (VLMs).** We use CLIP (or variants thereof) [56, 68] to perform fine-grained visual classification as a means to evaluate the usefulness of the generated descriptions. CLIP models are trained with contrastive image-text matching losses to rank matching image and text inputs highly and non-matching inputs lowly. [56, 68] show that CLIP variants generalize well to match unseen texts, e.g. class names, an ability commonly referred to as zero-shot classification.

First, the image to classify is converted into a normalized feature representation $I$ using CLIP's pre-trained vision backbone. Then, the class names are embedded into normalized feature vectors $T_N$ using the pre-trained text backbone. Next, all pairwise cosine similarities $I \cdot T_N$ of the respective feature representations are computed. Finally, the $n^* = \arg\max_N(I \cdot T_N)$ over these similarities reveals the most similar class $n^*$.

**Inference.** We generate a description $D_n^{(p)}$ with the above prompt for each class $n$ for each persona $p$ where we use a generative approach, i.e. we auto-regressively sample a random token from the predicted logits (see Figure 1, the green path). For Vicuna-13B we use the default temperature of 0.7 and the default top-k value of $k = 50$. For ChatGPT we use the default temperature of 1.0. This continues until the model emits an *<end of sequence>* or the maximum number of tokens (96) is reached. We did not tune these values.

For visual classification, we use the zero-shot classification capabilities of CLIP models, but instead of using the embedded class name itself ($T_n$), we use the embedding of the generated descriptions $D_n^{(p)}$ for each class $n$ and for each persona $p$. The predicted class for each persona $i^{(p)^*}$ is:

$$n^{(p)^*} = \arg\max(I \cdot D_n^{(p)}) \tag{4}$$

Performance is measured by computing the classification accuracy of the test splits on both datasets. As the descriptions are sampled from the LLM output, the results of the experiments are stochastic and we repeat them five times. We report the mean performance as well as 95% confidence intervals.

## 4 Experiments

Using Vicuna-13B, we evaluate the two-armed bandit and MMLU language reasoning tasks. For the zero-shot image classification task using a VLM we generate descriptions with both Vicuna-13B and ChatGPT. We focus on highlighting how the chosen persona changes the task performance of the LLM. As LLMs seem to be sensitive to prompts [69], we follow the meta-prompting approach from [26] to vary our impersonation prompts. We run all Vicuna-13B experiments with each of the six prompt variations, which are shown in the suppl. Section A.1. All experiments are performed on the test splits using a single A100-40GB GPU and we mention inference times in suppl. Section A.2.

### 4.1 Age-based impersonation changes exploration strategies

In the bandit task, for every age group that the LLM impersonates, we perform 2k two-armed bandit games of 10 trials each for each prompt variation. We evaluate the task performance in three ways.

First, we show the average reward per trial the LLM obtained with personas of increasing age in Figure 2 (top). With an increasing number of trials, the LLM obtains a higher average reward, corroborating that Vicuna-13B is able to learn from past trials to improve its policy similarly to GPT-3 in [8]. Moreover, as the LLM takes on a persona of different ages, we observe a divergence of obtained rewards as the number of trials increases. Younger personas, i.e., 2- and 4-year-old personas, obtain a smaller reward than older ones, i.e., 13- and 20-year-old personas.

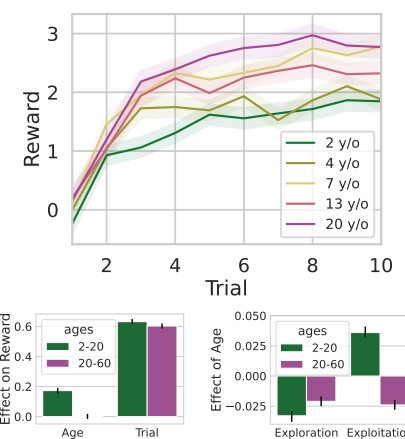

Secondly, we analyze the resulting rewards by using a regression, entering the trial number and age as independent variables. To extend the analysis, we evaluate two age groups, from 2 to 20 and from 20 to 60, where we evaluate ages in steps of 2 between 2 and 30 and steps of 5 from 30 to 60. We report these results in Figure 2 (bottom left). We find that the impersonating LLMs generally improved over trials, i.e. they increase their rewards as they progressed over trials of a game ($\beta = 0.63$, $p < .001$ for ages 2–20 and $\beta = 0.60$, $p < .001$ for ages 20–60). Importantly, LLMs impersonating older participants generate higher average rewards until age 20 ($\beta = 0.17$,

Figure 2: Two-armed bandit task. Top: Average reward per persona (10k games of 10 trials), left: Age and # of trials have a positive effect on the expected reward, right: With age, exploration decreases, and exploitation increases.

$p < .001$), thereby replicating a general pattern found in the developmental literature [70]. We find no significant effect from ages 20–60, which also mirrors observations of stagnating mental performance of adults.

Lastly, we analyze how regression weights of the probit-regression were influenced by the age group the LLM is impersonating, again analyzing ages 2–20 and 20–60. Figure 2 (bottom right) reveals that LLMs pretending to be older explored their environment less ($\beta = -0.03$, $p < .001$) and exploited more ($\beta = 0.04$, $p < .001$) in the ages between 2–20. This pattern is in line with several results from the psychological literature which also found that children show higher levels of directed exploration [71] than adults [72]. These results suggest that impersonating LLMs can recover human-like developmental stages of exploration in a two-armed bandit task. If life is seen as an exploration-exploitation problem, then younger agents should show higher amounts of directed exploration [73, 74]. To the best of our knowledge we are the first to show that LLMs replicate similar trends when using in-context impersonation.

### 4.2 Expertise-based impersonation changes reasoning abilities

Our experiments on expertise-based impersonation (details in Section 3.3) are conducted on the MMLU dataset [67], for which we ask Vicuna-13B to impersonate experts from three different categories (task, domain, and non-domain). For each task we compute the task accuracy averaged over all task questions (95% confidence intervals are computed over the average task accuracy). We compare the task expert results with the average of all domain expert personas, the average of all

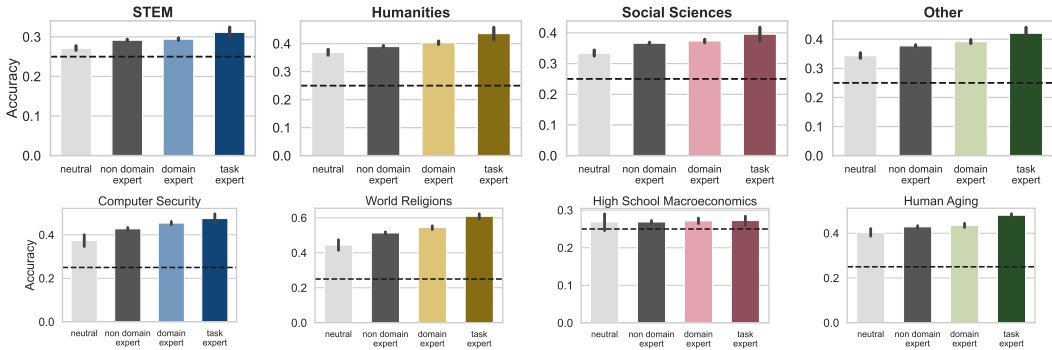

Figure 3: Expertise-based impersonation on all domains of the MMLU reasoning benchmark (top) and on exemplary individual tasks (bottom). For each task, we consider four personas: the neutral, the task expert, the domain experts (all experts from the same domain except the task expert) and the non-domain experts (all experts from all remaining domains). The dashed line is the random baseline.

non-domain expert personas, the average of all neutral personas, and the random baseline (horizontal line). We consider four neutral personas, namely student, average student, person, and average person, and the six aforementioned prompt variations.

In Figure 3 (top row), as expected, when the LLM is asked to impersonate the task expert, the performance is the highest. This shows that the LLM can indeed impersonate task experts with accuracy higher than random. Similarly, the domain expert personas perform better than the non-domain expert personas. This trend holds for all four MMLU domains and thus for MMLU in its entirety. In general, we observe that the performance in the Humanities tasks is higher than the accuracy in the other domain tasks, which is in line with results reported in the literature [61, 75, 76, 67]. Overall, these results suggest that LLMs can increase their performance when asked to impersonate task experts compared to non-task experts.

To provide more details on the individual behaviors of these personas, in the plots on the bottom row of Figure 3, we sample various expert personas, e.g. three positive and one negative case. The first, second and last plots indicate that the task expert persona performs better than the domain expert persona, which, in turn, outperforms the non-domain expert persona. In those cases, all experts outperform the neutral persona. For the High School Macroeconomics task, the task expert persona performs close to random and to the non-domain expert persona. This may be because, as Hendrycks et al. [67] observed, LLMs tend to perform worse on procedural problems that are calculation-heavy compared to purely verbal tasks. Furthermore, when the LLM performs close to or below the random baseline, i.e. the task is more difficult to solve for all types of experts, the impersonation trends are not as clear, since the model does not know how to solve the task well, irrespective of the persona. Thus, while in the Social Sciences field, the High School Macroeconomics task has worse performance, we see that for World Religions, the exam result is higher than 60%, i.e. a passing grade. Especially for World Religions and Human Aging, we observe that the task expert performs much better than the corresponding domain expert personas. We show results for all tasks in Section C.1 of the suppl.

Finally, since several MMLU evaluations [67, 77], can lead to small variations when comparing different models', we include results with the MMLU official prompt in suppl. Section C.2, where we verify that our findings on impersonation are not dependent on the formulation of the task. Lastly, we also show MMLU results for social groups in C.3.

### 4.3 Impersonation as categorical descriptions is complementary for visual categorization

In this section, we provide experimental results on two state-of-the-art fine-grained visual categorization datasets, i.e. Caltech UCSD Birds (CUB) [78] and Stanford Cars [79], with 200 and 196 classes of birds and cars, respectively. Additional results for FGVC Aircraft [80] and Oxford Flowers [81] can be found in Section D.2 of the supplementary. We first compare how different VLMs make use of the generated descriptions, then compare different LLMs in our in-context impersonation tasks and finally provide some qualitative results.

**Comparing VLM variants.** We first compare the classification accuracy of different VLMs when the Vicuna-13B generated descriptions of classes are fed to the language encoder of the VLM. For

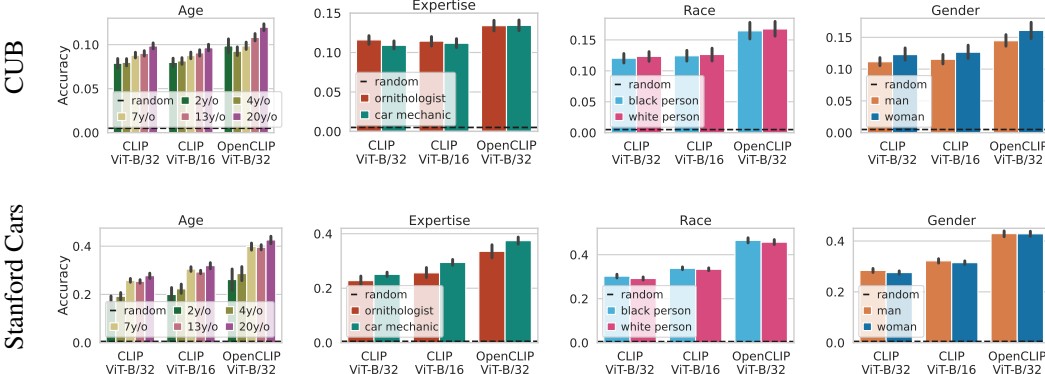

Figure 4: Comparing CLIP-32, CLIP-16 and OpenCLIP as VLMs (the language input comes from Vicuna-13B) on CUB (top) and Stanford Cars (bottom) datasets. We observe the effects of age, expertise, ethnicity and gender independent of the VLM used for fine-grained visual classification. The dashed line represents the random baseline.

the vision encoders we consider the Vision Transformer (ViT) [82] based B/32 and B/16 variants of the official CLIP implementation [56] as well as the OpenCLIP B/32 ViT variant [68]. The latter is a replication of the original CLIP trained on a larger dataset (Laion 5B [83]). For each CLIP variant, we use the corresponding causal transformer text encoders, which might not encode text as well as Vicuna but are able to embed the text into a shared multi-modal space.

Our results in Figure 4 show that across all three CLIP variants increased age in the impersonated persona increases performance for both bird and car classification. Interestingly, there is a significant increase in performance at 7 years of age when recognizing cars. Our expertise evaluation shows that the car mechanic persona's descriptions performs better than ornithologist's when recognizing cars. Interestingly, racial (column 3) and gender (column 4) personas, reveal consistent biases. While the black performs better in car classification, the white performs better in bird classification. This may indicate that there are stereotypical biases in the training data. Similarly, while the woman performs clearly better than man for bird classification, the trend is not as strong for car classification although man performs slightly better than woman. The language encoder of VLMs potentially being weaker than Vicuna, we expect these results to improve overall with a stronger language encoder in the VLM but this is an orthogonal direction to explore. To confirm the significance of our results, we run $Chi^2$ tests for expertise, race and gender. We consider the three CLIP models, five different seeds and the six different impersonation prompt variations. We find that for all experiments considered, {CUB, Stanford Cars} x {man/woman, black/white, ornithologist/car mechanic}, p<0.001. Thus, we conclude that our results are significant.

We also investigate the effects of composing personas for a computationally feasible subset of persons. More specifically, we study all possible combinations of {Black, White} × {Female, Male} for the CUB dataset for 5 different seeds (Figure 6). With Vicuna-13B we see weak evidence that the biases co-construct: Individually the white persona outperforms the black persona and the same applies to the female persona outperforming the male persona. Combined, the white female persona outperforms both the black female persona (change in race) and the white male persona (change in gender). Furthermore, we also study performance of additional genders (agender and non-binary) and races (indian, asian and hispanic) in the suppl. in Section D.5.

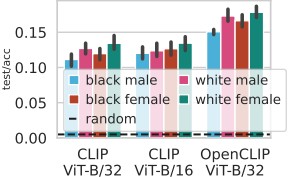

Figure 6: Composition of personas on CUB for Vicuna-13B.

**Comparing LLM variants** We evaluate how different LLMs, namely Vicuna-13B and ChatGPT, generate descriptions of the classes of interest. In these experiments, we keep the VLM fixed to OpenCLIP, as it is the best of the CLIP variants tested above. For computational reasons, we only evaluate on our original impersonation prompt. Figure 5 shows the effect of LLM impersonation on the generated descriptions evaluated on zero-shot image classification.

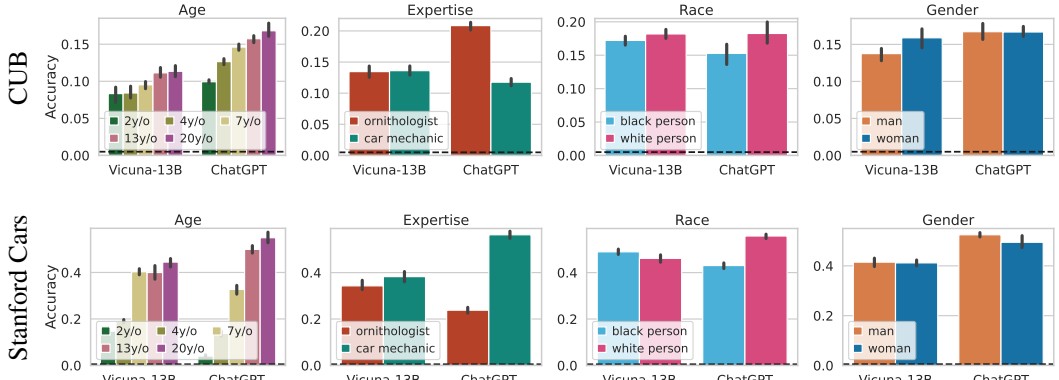

Figure 5: Comparing Vicuna-13B and ChatGPT as LLM variants (OpenCLIP is the VLM) on CUB and Stanford Cars. For both LLMs, the accuracy increases with increasing age, the expert persona on the respective dataset performs better and both LLMs are not free of biases, and impersonation of different genders or race affects their performance. The dashed line represents the random baseline.

For the age personas, we observe a clear trend of increased performance for both LLMs as they impersonate older characters. The progression is particularly pronounced for ChatGPT, where on Stanford Cars the 2-year-old persona describes different cars with similar expressions leading to $\sim 4\%$ accuracy, but as ChatGPT's persona gets older, it becomes more accurate in describing cars, e.g. 54.9% for persona of age 20. This indicates that LLMs can replicate human language at different development stages, varying their language both in terms of vocabulary and general knowledge for accurately describing these objects as discussed in [84]. Similarly to the reasoning task, LLMs exhibit higher expertise on the topic when we ask them to impersonate a bird expert ("ornithologist" persona) and a car expert ("car mechanic" persona). The respective domain expert persona performs approximately twice as well as the non-domain expert persona when using ChatGPT. Impersonating an expert, the LLM tends to describe a class in more detail and mention more discriminative features.

We also observe that impersonation can reveal biases encoded in the LLMs. A race bias becomes apparent when we ask the LLMs to impersonate a "black" or "white" person. ChatGPT tends to describe both birds and cars better when posing as a white person. Vicuna-13B, on the other hand, provides better descriptions of cars as a black person. Gender biases are a bit less noticeable, but we still find Vicuna-13B giving better bird descriptions as a woman persona and ChatGPT identifying cars better as a man persona. While instruction-based fine-tuning [64] tries to remedy social biases encoded in LLMs to some extent, we can still expose them through in-context impersonation.

Overall, we find that ChatGPT shows larger effects, probably due to its access to more diverse (fine-tuning) data. The fact that the effects described above can be found with two very different language models suggests that they are a result of the overall language modeling and instruction following training on internet data instead of specific model artifacts.

**Qualitative results and limitations.** In Figure 7, we provide the descriptions generated by ChatGPT and Vicuna for one class, i.e. black billed cuckoo, from the CUB dataset and one class, i.e. AM General Hummer SUV 2000, from the Stanford Cars dataset. As personas, we sample all the age personas we considered in our experiments, namely 2, 4, 7, 13 and 20-year-old personas.

For both LLMs, in both datasets, we observe that with increasing age, the complexity of the vocabulary and attributes of the mentioned objects increases. A 2-year-old persona talks about the sound the bird or the car makes, the shapes of the wings or wheels, and the emotions attached to seeing or riding it. A 4-year-old persona interestingly mentions experiences seeing the bird or the car more distinctly. A 7-year-old persona starts using more complicated adjective phrases, e.g. can drive on rough roads and outside places, whereas a 13-year-old persona takes it one step further, e.g. brownish-gray body with distinctive rusty colored markings. Finally, a 20-year-old persona makes a more complete description of the object including where the bird is found or what the car is mainly used for. This is in line with [85] where the authors show that given the same length of text, smaller children use less diverse and non-academic vocabulary, and repeat a lot. Even though LLM's may not faithfully represent the language of children, we qualitatively observe similar patterns. We show more examples and quantize the properties of the generated descriptions in suppl. Section D.3.

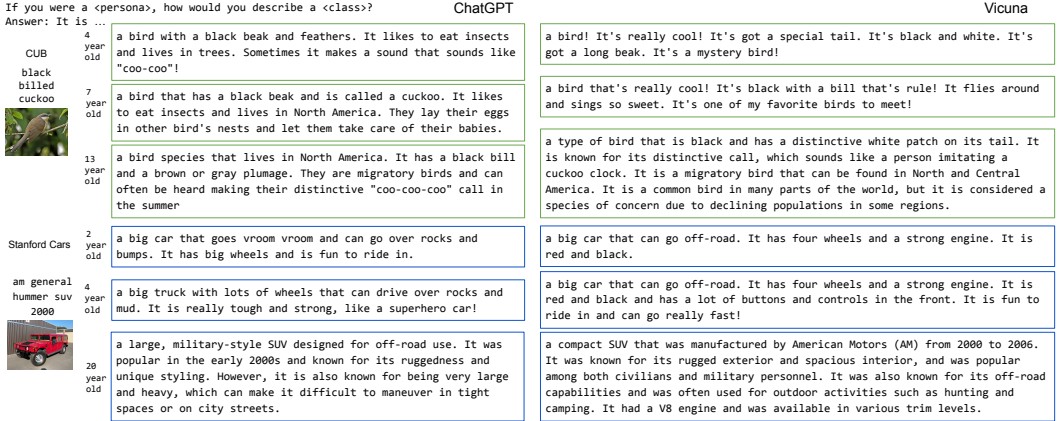

Figure 7: Qualitative results sampling all the age personas (2, 4, 7, 13 and 20-year-old personas) for two classes, i.e. Black Billed Cuckoo (CUB) and AM General Hummer SUV 2000 (Stanford Cars) classes. The results are obtained by querying ChatGPT and Vicuna.

One obvious difference between these two LLMs to point out is that the descriptions obtained from Vicuna appear to be longer and more detailed. Further, at earlier ages, e.g. 2 or 4, especially on CUB, the descriptions of Vicuna seem poetic. The difference between the semantic content of the descriptions of the 13-year-old persona and the 20-year-old persona seems to be less distinct in Vicuna than in ChatGPT. One final interesting observation is that Vicuna descriptions talk about the color of the car whereas the color can not be a distinguishing property of a car.

## 5    Broader Impact

We believe that a better understanding of in-context impersonation, as well as its resulting downstream effects, can not only help to mitigate the risk of fraud but also to understand how these newly-powerful agents behave more generally [86]. We have already seen that in-context impersonation boosts performance and produces biases; these results could be followed up by investigating how these characteristics emerge during training, change with increasing model size [87], or adapt with additional fine-tuning [88]. Additionally, LLM providers could quantitatively test for these biases before releasing new models. We specifically discourage crafting (system) prompts for maximal performance by exploiting biases, as this may have unexpected side effects, reinforce societal biases and poison training data obtained with such prompts. Other misuses may include amplification of stereotypical biases through generated content and using impersonation to invoke fake trust. However, we believe systematically studying these biases raises awareness in the ML community and general society and serves as a first step to research mitigation strategies. Lastly, we discuss limitations of our work in suppl. Section E.

## 6    Conclusion

We presented evidence that *in-context impersonation*, that is asking LLMs to take on different roles in context, can change their performance and reveal their biases. Asking LLMs to impersonate differently aged people in a two-armed bandit task, LLMs could reproduce human-like developmental stages of exploration behavior. Asking LLMs to impersonate domain experts, they performed better than LLMs that were asked to impersonate a non-domain expert. Finally, asking LLMs to impersonate various roles in a vision-language task revealed not only that impersonation can boost relative performance but also recovered societal biases about a person's age, gender, and race.

We have demonstrated the effects of in-context impersonation on single agents performing relatively simple tasks across a limited range of personas. In future work, we want to scale up this approach to multiple LLMs impersonating a variety of personas across complex and interactive tasks [89]. Finally, we believe that in-context impersonation can also be applied to other modalities, for example to large models for video generation [90].

# 7 Acknowledgements

The authors thank IMPRS-IS for supporting Leonard Salewski. This work was partially funded by the Portuguese Foundation for Science and Technology (FCT) under PhD grant 2020.07034.BD, the Max Planck Society, the Volkswagen Foundation, the BMBF Tübingen AI Center (FKZ: 01IS18039A), DFG (EXC number 2064/1 – Project number 390727645) and ERC (853489-DEXIM).

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
