# Supplementary Materials: In-Context Impersonation Reveals Large Language Models' Strengths and Biases

**Leonard Salewski**[1,2]    **Stephan Alaniz**[1,2]    **Isabel Rio-Torto**[3,4*]

**Eric Schulz**[2,5]    **Zeynep Akata**[1,2]

[1] University of Tübingen    [2] Tübingen AI Center    [3] University of Porto
[4] INESC TEC    [5] Max Planck Institute for Biological Cybernetics

In this supplementary materials we show additional results mentioned in the main paper. First, we give experimental details in Section A. Next, we show results for Llama 2 on the bandit task in Section B. Afterwards, we show in Section C.1 additional quantitative results for the expertise-based impersonation results. Section D provides additional details about the vision and language tasks. Finally, Section E discusses some limitations of our work.

The code to reproduce our results is available here: `https://github.com/ExplainableML/in-context-impersonation`. For more details on the code please refer to the `README.md` file.

## A    Experimental Details

This section describes the meta-prompt that we used to generate the prompt variations (Section Section A.1) and the amount of compute required to reproduce our experiments (Section Section A.2)

### A.1    Prompt variations generated by meta-prompting

As LLMs have been found to be sensitive to specific prompts [1], we follow the meta-prompting approach from [2] to vary our original impersonation prompt `If you were a {persona}`. We generated five additional variations with the following meta-prompt:

```
Write 5 different grammatical and linguistic variations of the following
        instruction.  You shall not fill in the curly brackets:
                        If you were a {persona}
```

The following enumeration lists all generated prompts, which we used in addition to the original prompt:

- `Should you be transformed into a {persona}`

- `Imagine you are a {persona}`

- `Should you assume the role of a {persona}`

- `Were you to take on the persona of a {persona}`

- `In the case of you being a {persona}`

---

*Work done whilst visiting University of Tübingen

37th Conference on Neural Information Processing Systems (NeurIPS 2023).

### A.2 Compute and Reproduction

For all Vicuna-13B based experiments (bandit, reasoning and vision) we used a single Nvidia A100-40GB GPU. The weights for this language model can be obtained from its open source documentation[2], making our Vicuna-13B based experiments fully reproducible. For our ChatGPT vision experiments we used Nvidia 2080ti 11GB GPUs to run the CLIP models.

For the bandit task, we chose to run 12k games (2k per prompt variation) with Vicuna-13B to obtain a large sample size for our analysis. Trials and games were run sequentially for approximately 3.5 hours per persona. Processing games and personas in parallel through batching could reduce the time needed for this experiment significantly.

For the Vicuna-13B reasoning experiments, running (sequentially) all 57 tasks and personas considered takes about 12 hours for a single prompt variation.

For the Vicuna-13B vision and language experiments, generating the descriptions for a single persona and for 196 (Stanford Cars) or 200 (CUB) classes and running CLIP zero shot classification with them on the entire test splits takes approximately an hour for a single impersonation prompt.

## B Bandit Task — Results for Llama 2

Most open-source models such as Vicuna are fine-tuned from the same base model Llama [3]. Recently, a new foundational open-source model, Llama 2 (70B, Chat variant) [4] has been released which is significantly larger than Vicuna-13B and has been trained on more data. We rerun the bandit experiments using Llama 2 and come to the same conclusions. The effect of age in the range of 2–20 on the reward is $\beta = 0.17$ ($p < .001$) and $\beta = 0.26$ ($p < .001$) for Vicuna-13B and Llama 2, respectively.

## C Reasoning Task

This section describes additional results regarding the MMLU reasoning task. We start by complementing the results of the main paper by presenting all 57 individual task plots in Section C.1. We then present a comparison between our prompt and the official MMLU prompt [5] in Section C.2 and, lastly, present results on race and gender social categories in Section C.3. All experiments are conducted on both Vicuna-13B and ChatGPT.

### C.1 Additional quantitative results for expertise-based impersonation

In the main paper, only a part of the Vicuna [6] related results were included for the MMLU [5] reasoning task, for which the LLM is prompted with a question and four answer options. Thus, in this section, we simultaneously provide the Vicuna-13B individual results for all 57 tasks considered, and a comparison with ChatGPT. These experiments are the result of the six prompt variations described in Section A.1.

Contrary to Vicuna, which is an open source model, ChatGPT does not offer direct access to the token probabilities. Therefore, we add the following expression to the Vicuna prompt mentioned in the main paper `Answer: The answer is option`, in order to force ChatGPT to provide one of the 4 options as the first generated token. This generated token is then taken as the LLM prediction. When ChatGPT does not provide one of the options as the first token, we repeat the question until a valid option is generated or until a maximum of 10 tries. If none of these conditions are met, we discard the sample. For example, for the STEM and Humanities domains, in about 250k questions (7835 unique questions, each of which evaluated for the 32 personas of these domains), only 178 were discarded (0.07%).

The aforementioned results are presented in Figures 1, 2, 3, and 4, for the STEM, Humanities, Social Sciences, and Other domains, respectively. ChatGPT performs consistently better than Vicuna-13B, which is also in line with the expectation given that ChatGPT is a larger model trained on more and higher quality (human feedback) data. Furthermore, as discussed in the main paper for Vicuna and again observed for ChatGPT, the performance on Humanities tasks is consistently higher than

---

[2]`https://github.com/lm-sys/FastChat`

on STEM tasks, which aligns with previous literature. For Vicuna-13B, the tasks where the trend is not verified (i.e. where the task expert does not outperform the domain expert and/or where the domain expert does not surpass the non-domain expert), coincide with tasks that the model could not perform well in general, i.e. had accuracies close to or below the random baseline for all personas considered (see Formal Logic in Figure 2 or, for example, College Chemistry, College Computer Science, High School Statistics). For ChatGPT, the tasks where the trend is not as clear coincide with tasks where Vicuna also had worse results. Interestingly, the neutral persona performs on par with the domain expert. Additionally, for the Other domain, ChatGPTs' expertise trends are not as clear, which might be due to the fact that this domain includes tasks from a very wide range of domains, such as Nutrition and Business Ethics, for example. Nevertheless, the non-domain expert is outperformed by the domain expert, who in turn is outperformed by the task expert for all four domains.

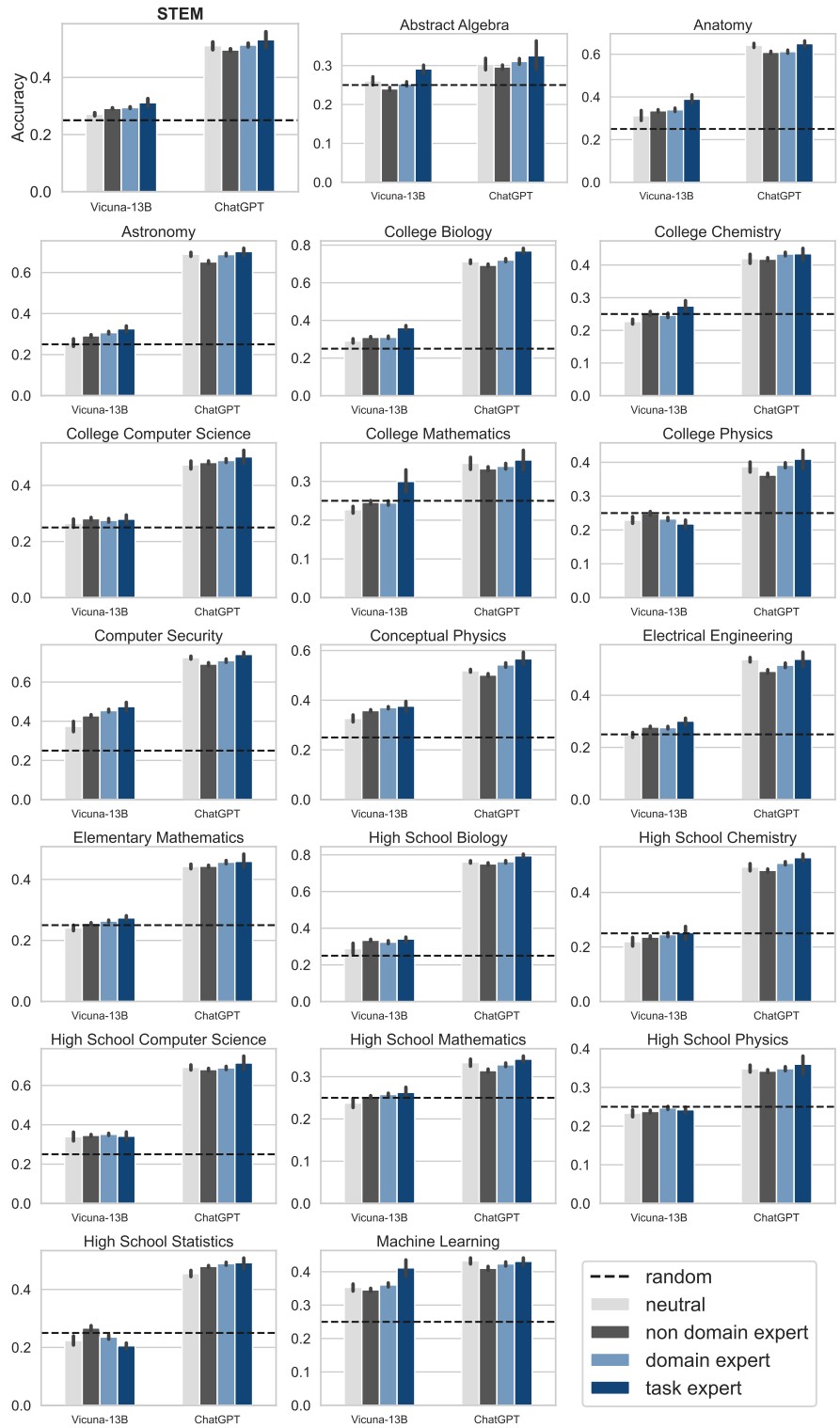

Figure 1: Comparison between Vicuna-13B and ChatGPT for expertise-based impersonation on the STEM domain of the MMLU reasoning benchmark. We compare the task expert results with the average of all neutral personas, the average of all domain expert personas, the average of all non-domain expert personas and the random baseline (horizontal line). The first plot shows the average over all STEM tasks, while the remaining plots show the results for each STEM task individually. All 95% confidence intervals are computed over the average task accuracy.

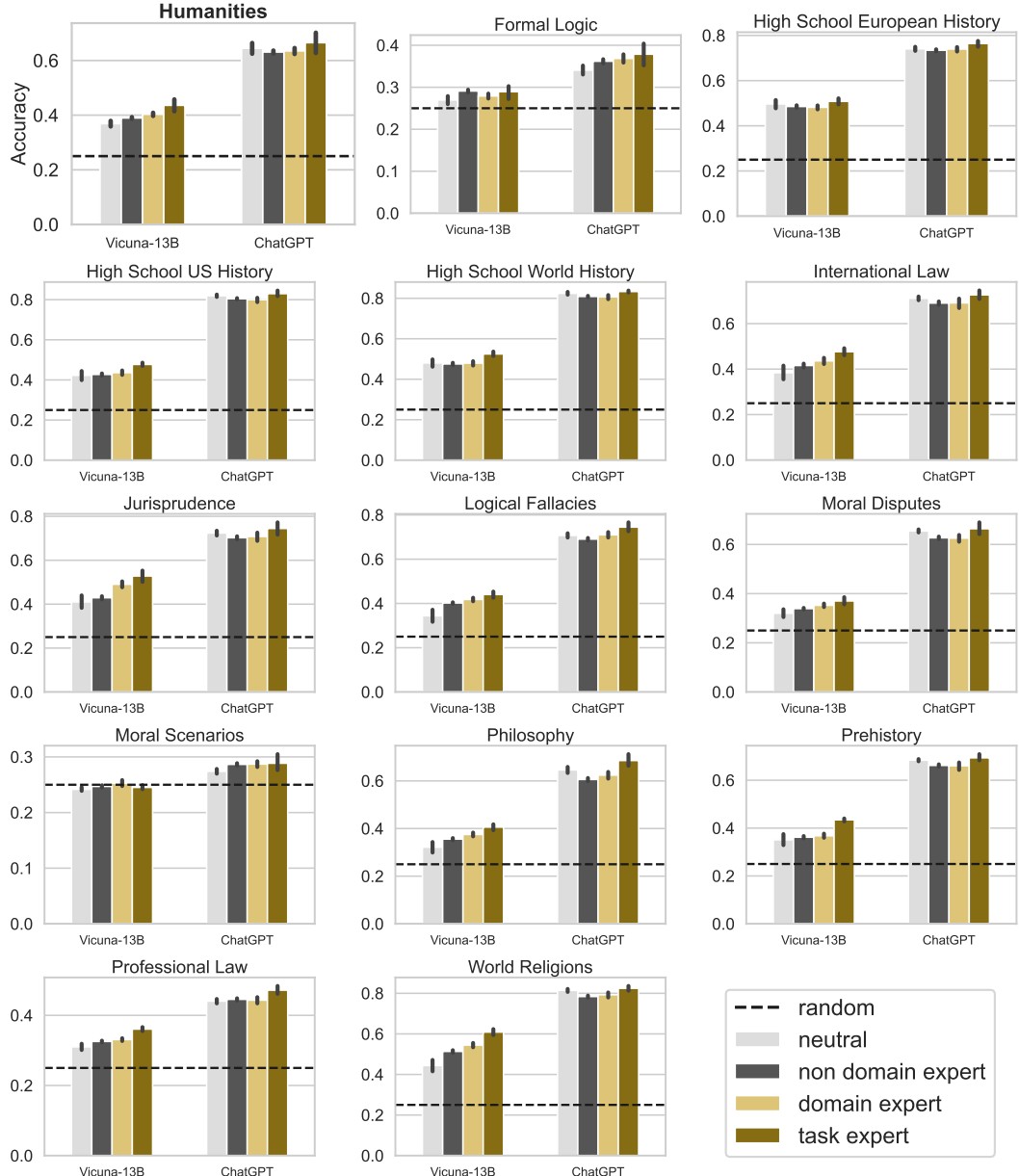

Figure 2: Comparison between Vicuna-13B and ChatGPT for expertise-based impersonation on the Humanities domain of the MMLU reasoning benchmark. We compare the task expert results with the average of all neutral personas, the average of all domain expert personas, the average of all non-domain expert personas and the random baseline (horizontal line). The first plot shows the average over all Humanities tasks, while the remaining plots show the results for each Humanities task individually. All 95% confidence intervals are computed over the average task accuracy.

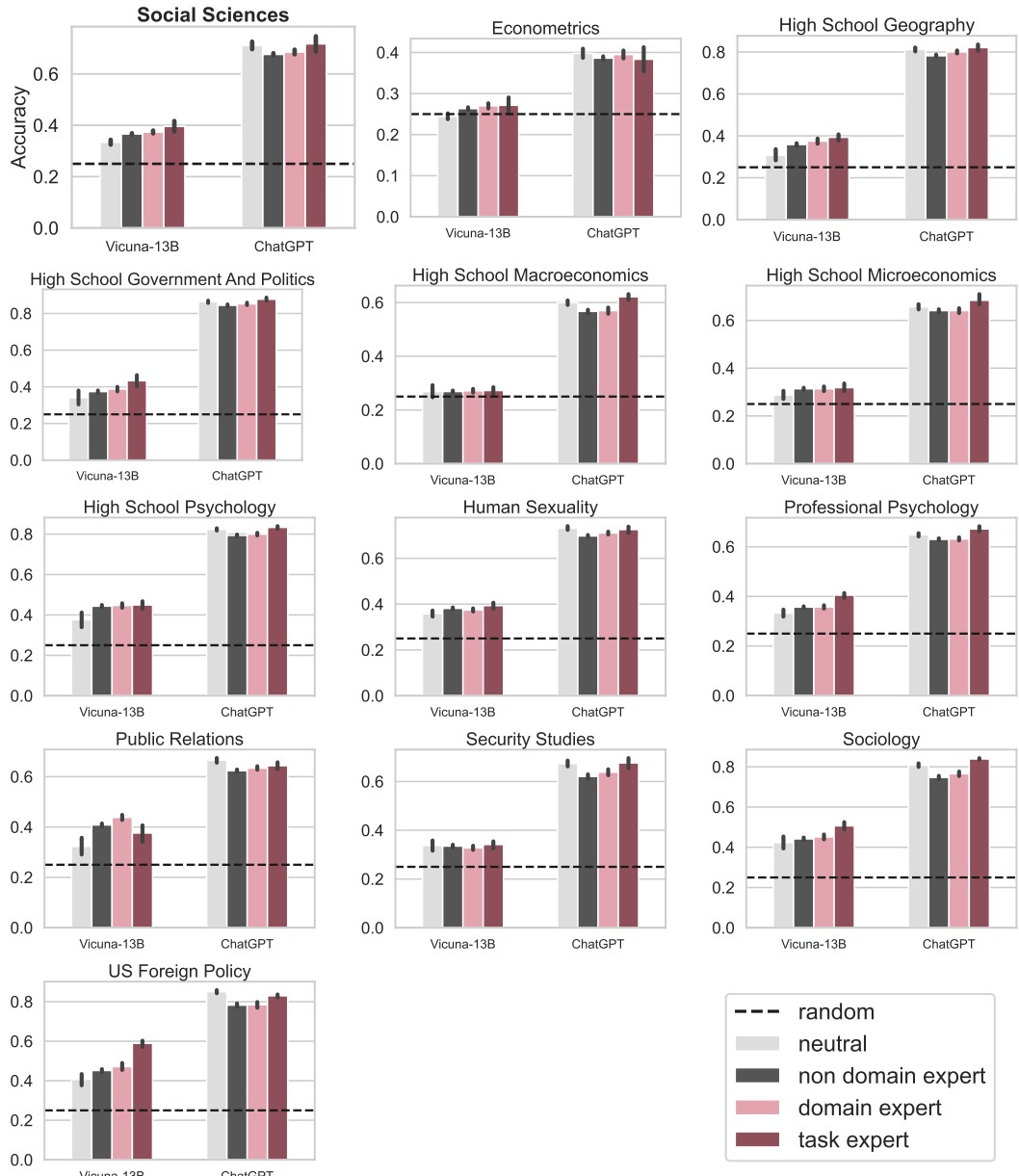

Figure 3: Comparison between Vicuna-13B and ChatGPT for expertise-based impersonation on the Social Sciences domain of the MMLU reasoning benchmark. We compare the task expert results with the average of all neutral personas, the average of all domain expert personas, the average of all non-domain expert personas and the random baseline (horizontal line). The first plot shows the average over all Social Sciences tasks, while the remaining plots show the results for each Social Sciences task individually. All 95% confidence intervals are computed over the average task accuracy.

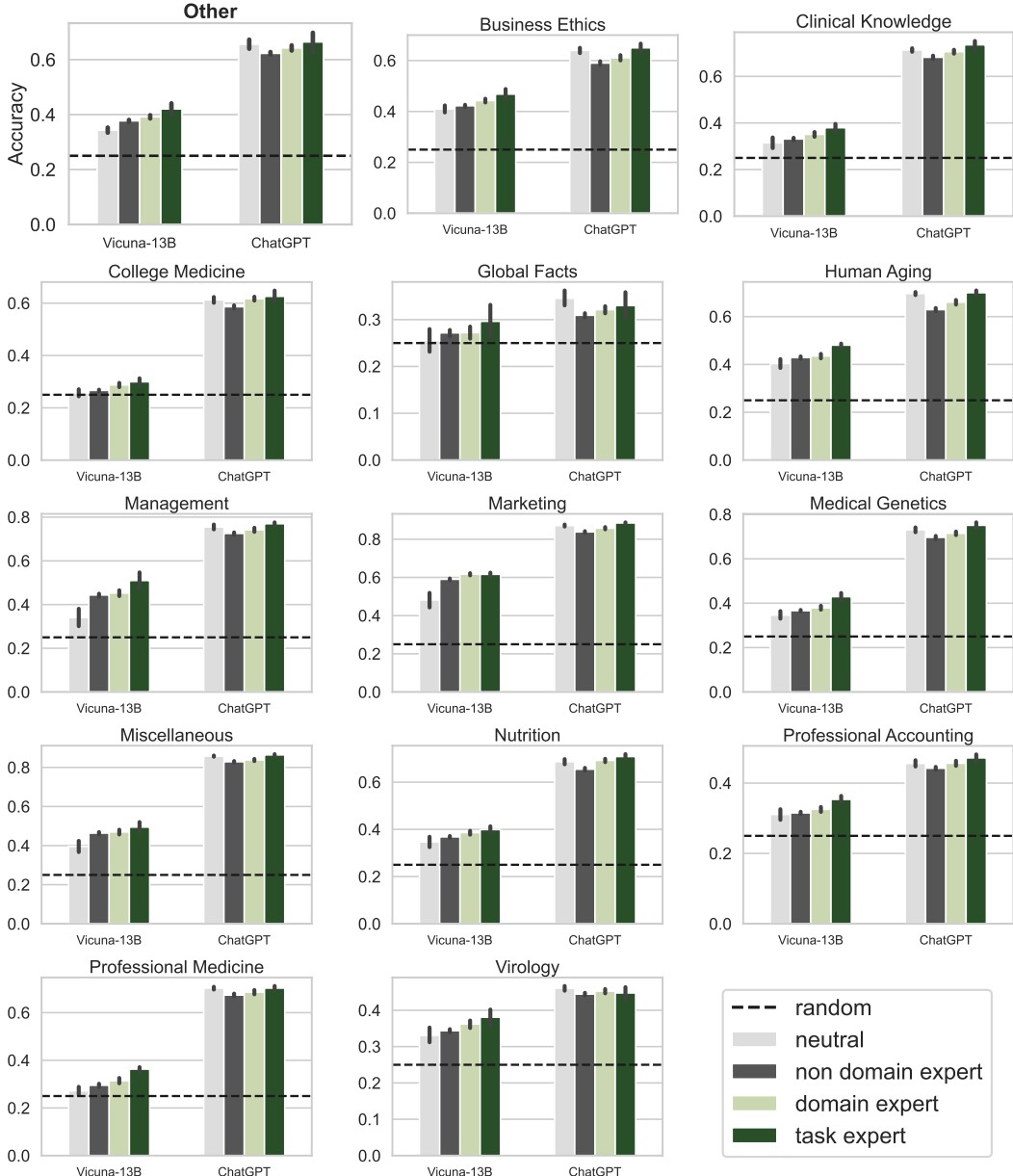

Figure 4: Comparison between Vicuna-13B and ChatGPT for expertise-based impersonation on the Other domain of the MMLU reasoning benchmark. We compare the task expert results with the average of all neutral personas, the average of all domain expert personas, the average of all non-domain expert personas and the random baseline (horizontal line). The first plot shows the average over all Other tasks, while the remaining plots show the results for each Other task individually. All 95% confidence intervals are computed over the average task accuracy.

## C.2 MMLU Task Formulation

Since several MMLU evaluations [5, 7], may lead to small variations when comparing different models' ranks, we include results with the MMLU official prompt (see Figure 5), i.e. by using the MMLU prompt at the start and keeping our impersonation strategy. Our expertise-based impersonation trends still hold, and the absolute accuracy values improve. This increase in accuracy might be explained by the fact that the official MMLU prompt includes the task name in the prompt, which might provide additional clues to the LLM. Thus, we conclude that our findings on impersonation are not dependent on the formulation of the task.

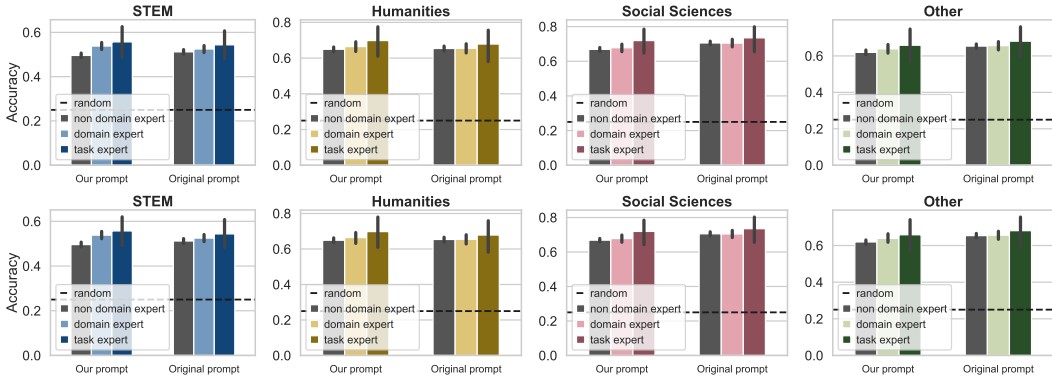

Figure 5: Comparison between our task formulation (Our prompt) and the official MMLU prompt [5] (Original prompt), for Vicuna-13B (top) and ChatGPT (bottom).

## C.3 Social Categories on MMLU

We present in Figure 6 results for both Vicuna-13B (top) and ChatGPT (bottom) on MMLU when considering different social category prefixes (black, white, male, and female). We observe that, for both models, the performance when impersonating experts while adding these prefixes is consistently lower than when no prefix is added (i.e. the none columns). For Vicuna, the black persona obtains lower accuracies than the white persona, especially regarding the non-task experts, and a female expert outperforms a male expert. For ChatGPT, all prefixed personas' performance is similar.

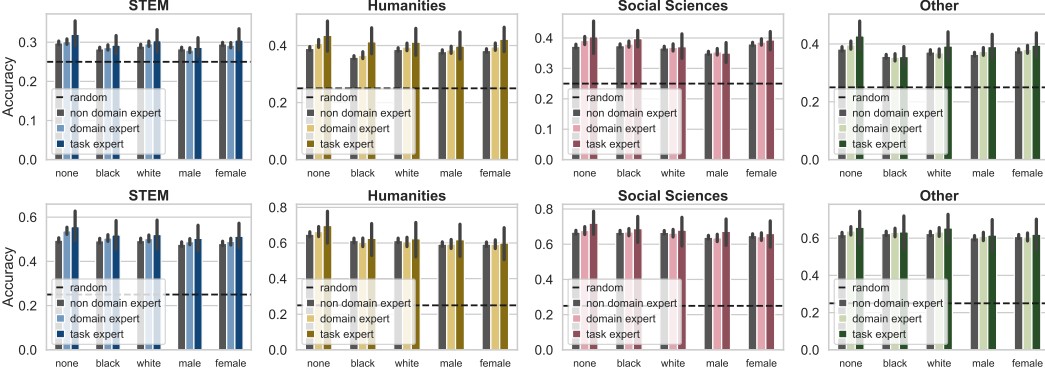

Figure 6: Expertise-based impersonation results with social category prefixes (black, white, male, and female) for Vicuna-13B (top) and ChatGPT (bottom).

# D Vision and Language Task

In this section we give additional details for the vision and language task. First, in Section D.1 we describe how class names were removed from the generated visual descriptions to avoid trivial solutions. Then we show more results on two additional fine-grained visual classification datasets in Section D.2. Next, we show more qualitative examples of the class descriptions generated by Vicuna-13B and ChatGPT in Section D.3. Afterwards, we show more quantitative results on more

LLM / VLM pairs in Section D.4. Lastly, we show more results for additional races and genders (Section D.5) and for Google PaLM (Section D.6).

## D.1 Removing class names from visual descriptions

When the class name is included inside the generated description, it has a significant effect on the downstream performance of the vision task. In such cases CLIP can classify images well without the need of additional descriptions. We find that both language models occasionally use the class name in their output. For example at the beginning of a new sentence. To actually measure how well a persona describes a class, we use a two step process to remove the class name from the descriptions.

**Manual cleaning.** We use a set of heuristics to remove the class name, e.g. replacing `A {class name} {verb}` with `It {verb}`. These heuristics account for the numerous (singular or plural) of the class name as well as for lower and upper casing variants. Whilst this approach is very fast, it does not scale to all possible variants how the class name could be mentioned in the generated descriptions.

**LLM based cleaning.** For the LLM based cleaning we first split the descriptions into individual sentences with spacy [8]. This simplifies the task for the LLM. To remove the class name in more complex settings we prompt the

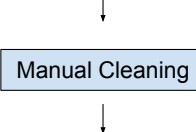

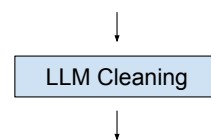

Figure 7: Example of the two step class name removal process.

same LLM used for generating the descriptions with four in-context examples. Empirically, we find this cleaning approach works well and can also handle more complex cases, e.g. removing parts of a sentence if needed. An example of this is shown in Figure 7. Lastly, if the result still contains the class name we use the original sentence, to avoid introducing any malformed LLM output.

## D.2 Additional visual datasets

We extend our analysis to other datasets and more categories by using FGVC Aircraft [9] (100 categories of aircraft from different manufacturers and eras) and Oxford Flowers [10] (102 categories of flowers with large scale, pose and light variations). For gender, we find significant performance differences when evaluating the descriptions generated by Vicuna-13B on the two additional datasets, strengthening our original argument that these LLMs exhibit biases (Figure 8). This means that descriptions generated by female personas outperform those generated by male personas across all three tested VLMs. For racial biases, we see only smaller differences across evaluation with different VLMs. The same trends hold for ChatGPT.

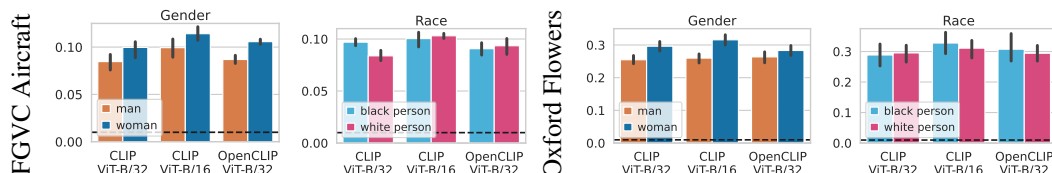

Figure 8: Evaluating bias of Vicuna-13B on more object categories (FGVC Aircraft [9] and Oxford Flowers [10]). The dashed line is the random baseline.

## D.3 Example descriptions for the visual classification task

Recall that for the visual classification task we ask the impersonating language models to generate a description for each class of the dataset. In the main paper we showed descriptions for two examples of a subset of all personas considered (4-, 7- and 13-year-old for CUB [11] and 2-, 4- and 20-year-old

for Stanford Cars [12]). In this section we show and discuss example descriptions on both vision datasets (CUB and Stanford Cars) generated to for all age ranges included in our paper (2-, 4-, 7-, 13- and 20-year-old).

The examples in Figure 9 support our findings from the main paper, that with increasing age of the persona the complexity w.r.t. e.g. vocabulary increases. For CUB we additionally show examples for the 2- and 20-year-old's and the differences in the wording are very apparent. For both language models the descriptions generated for the 2-year-old are short and have simple grammatical structures. In contrast, for the 20-year-old the descriptions exhaust much more of the 45 word instruction and use words that are not part of the vocabulary of a 2-year-old (e.g. `migratory bird` or `protected species`).

For Stanford Cars we additionally include the descriptions generated for the 7- and 13-year-old personas. In contrast to the 4-year-old both descriptions are much longer, including many facts about the cars (e.g. the manufacturer of the car).

### D.3.1 Analysis of text complexity for different age groups

In Figure 6 of the main paper as well as in Figure 9 we qualitatively described how the text changes as we vary the age of the impersonated person. To understand how the generated descriptions quantitatively change we also evaluate the complexity of the generated descriptions.

We use the `textstat` package[3], which runs several different text complexity metrics [13–19] and creates an aggregate consensus score that indicates which grade in school is at least required to read the texts. In Figure 10 we show the results for both, CUB and Stanford Cars.

We find, that across all language models and both datasets the impersonation of differently aged personas increases the required grade level to read the descriptions. For CUB the grade level increases not as much (from 4th to 9th grade) than on Stanford Cars (from 3rd to approx. 10th grade). This might be due to the fact that more descriptions of the oldest personas mention complex terms like manufacturers for the Stanford Cars dataset.

### D.4 Quantitative results on LLM / VLM pairs

In Section 4.3, Figure 4 of the main paper, we show results for the three different CLIP variants (CLIP with ViT B/32, ViT B/16 and OpenCLIP [20]) based on the descriptions generated by the Vicuna-13B LLM. Here we additionally show these results for descriptions generated with ChatGPT in Figure 11.

Similar to the findings on Vicuna-13B the descriptions generated by ChatGPT exhibit an increase in fine-grained visual classification performance as the age of the impersonated person increases. For ChatGPT this effect is more clear on Stanford Cars than on CUB. Additionally, these results confirm our finding that expert impersonations perform better than non-experts. However, for ChatGPT the effect is even more clear; the expert performs roughly twice as well as the non-expert across all VLMs. Regarding race, ChatGPT descriptions' seem to have more bias than those generated by Vicuna. Lastly, for different genders, we find ChatGPTs' descriptions of female impersonation to perform consistently worse than those of male impersonation.

### D.5 Additional bias groups

We also study additional bias groups for race and gender in Figure 12. For gender we study agender and non-binary and for race we study indian person, asian person and hispanic person. For CUB we find for gender that performance of the agender and non-binary personas is similar to the performance of the female persona. On Stanford Cars the non-binary persona outperforms the agender persona. For race we find overall worse performance compared to the white and black personas. Overall we conclude that biases can also be found in additional bias groups.

### D.6 Results for Google PaLM

Additionally, we ran exploratory experiments using the proprietary Google PaLM model [21, 22] via their API. When we apply in-context impersonation of racial biases, the API does not provide

---

[3]`https://github.com/textstat/textstat`

If you were a <persona>, how would you describe a <class>?
Answer: It is ...

ChatGPT

**CUB**
black billed cuckoo

| | |
|---|---|
| 2 y/o | a bird that goes "cuckoo". It has a black bill and likes to eat bugs. It can fly and live in trees. It makes a funny sound that makes me laugh. |
| 4 y/o | a bird with a black beak and feathers. It likes to eat insects and lives in trees. Sometimes it makes a sound that sounds like "coo-coo"! |
| 7 y/o | a bird that has a black beak and is called a cuckoo. It likes to eat insects and lives in North America. They lay their eggs in other bird's nests and let them take care of their babies. |
| 13 y/o | a bird species that lives in North America. It has a black bill and a brown or gray plumage. They are migratory birds and can often be heard making their distinctive "coo-coo-coo" call in the summer |
| 20 y/o | a bird species found in North America. It has a distinct black bill and is known for laying its eggs in other bird species' nests. Often heard more than seen, it has a unique call that sounds like a slow, hollow wooden "cuckoo." |

**Stanford Cars**
am general hummer suv 2000

| | |
|---|---|
| 2 y/o | a big car that goes vroom vroom and can go over rocks and bumps. It has big wheels and is fun to ride in. |
| 4 y/o | a big truck with lots of wheels that can drive over rocks and mud. It is really tough and strong, like a superhero car! |
| 7 y/o | a big and strong car that can drive on rough roads and outside places. It looks like a military vehicle and can hold lots of people inside. It is really cool! |
| 13 y/o | a really cool and tough-looking SUV that was made by AM General in the year 2000. It's known for being able to go off-road and handle all kinds of terrain. |
| 20 y/o | a large, military-style SUV designed for off-road use. It was popular in the early 2000s and known for its ruggedness and unique styling. However, it is also known for being very large and heavy, which can make it difficult to maneuver in tight spaces or on city streets. |

If you were a <persona>, how would you describe a <class>?
Answer: It is ...

Vicuna

**CUB**
black billed cuckoo

| | |
|---|---|
| 2 y/o | A bird that is black and has a curved bill. |
| 4 y/o | It's a bird! It's really cool! It's got a special tail. It's black and white.It's got a long beak. It's a mystery bird! |
| 7 y/o | It's a bird that's really cool! It's black with a bill that's rule! It flies around and sings so sweet. It's one of my favorite birds to meet! |
| 13 y/o | It is a type of bird that is black and has a distinctive white patch on its tail. It is known for its distinctive call, which sounds like a person imitating a cuckoo clock. It is a migratory bird that can be found in North and Central America. It is a common bird in many parts of the world, but it is considered a species of concern due to declining populations in some regions. |
| 20 y/o | A bird species that is common in parts of North and South America, known for its distinctive black bill and brown-gray feathers. It is a migratory bird, and is often seen in wooded areas near rivers and marshes. The Black-billed Cuckoo is known for its unique call, which sounds like a repeated "coo-coo." It is a protected species and is considered an important part of the ecosystem. |

**Stanford Cars**
am general hummer suv 2000

| | |
|---|---|
| 2 y/o | It is a big car that can go off-road. It has four wheels and a strong engine. It is red and black. |
| 4 y/o | It is a big car that can go off-road. It has four wheels and a strong engine. It is red and black and has a lot of buttons and controls in the front. It is fun to ride in and can go really fast! |
| 7 y/o | It is a type of car that is like a big truck but still has four wheels and can drive on roads like a regular car. It is made by a company called AM General and was first made in the year 2000. It is pretty big and can hold a lot of people or things inside. It is often used for driving in rough or off-road environments. |
| 13 y/o | It is a type of sport utility vehicle (SUV) that was manufactured by the American automaker AM General in the year 2000. It is known for its rugged appearance and off-road capabilities. The Hummer SUV was popular in the early 2000s, but production of the vehicle stopped in 2010 due to declining sales and environmental concerns. |
| 20 y/o | It is a compact SUV that was manufactured by American Motors (AM) from 2000 to 2006. It was known for its rugged exterior and spacious interior, and was popular among both civilians and military personnel. It was also known for its off-road capabilities and was often used for outdoor activities such as hunting and camping. It had a V8 engine and was available in various trim levels. |

Figure 9: Qualitative results for all the age personas (2, 4, 7, 13 and 20-year-old personas) for two classes, i.e. Black Billed Cuckoo (CUB) and AM General Hummer SUV 2000 (Stanford Cars) classes. The results are obtained by querying ChatGPT and Vicuna.

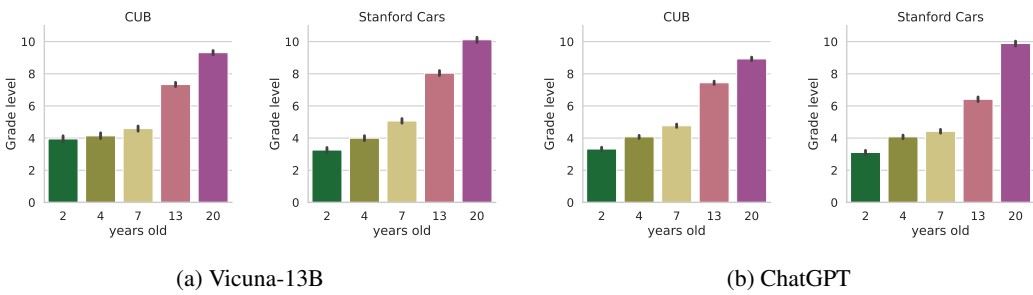

(a) Vicuna-13B                                                    (b) ChatGPT

Figure 10: Text complexity on Vicuna-13B (left) and ChatGPT (right) for CUB and Stanford Cars.

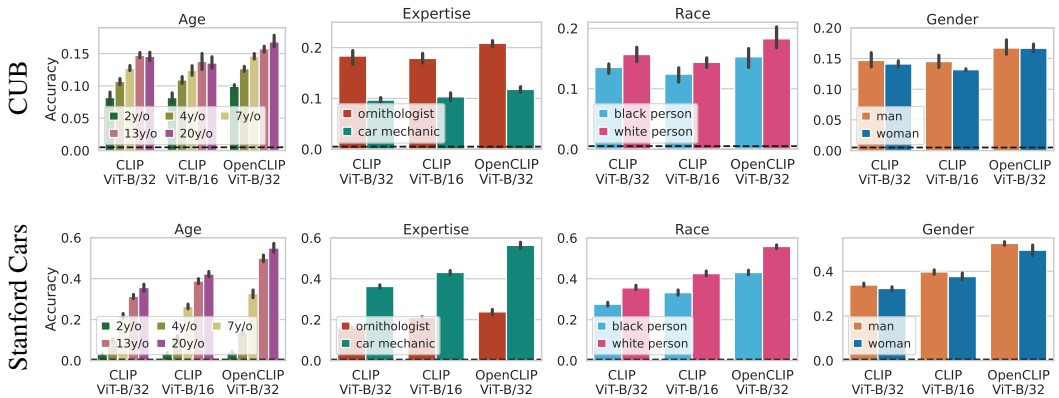

Figure 11: Comparing CLIP-32, CLIP-16 and OpenCLIP as VLMs. In contrast to Figure 4 in the paper (which shows Vicuna-13B results), the language input here comes from ChatGPT. We observe the effects of age, expertise, race and gender independent of the VLM used for fine-grained visual classification on the CUB (top) and Stanford Cars (bottom) datasets. The dashed line is the random baseline.

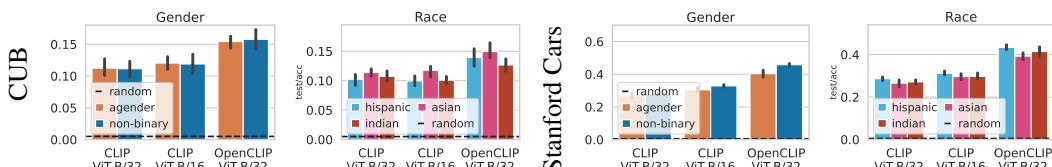

Figure 12: Evaluating more genders and races for Vicuna-13B on CUB and Stanford Cars.

an answer because the LLM model output is flagged by a text classifier to be unsafe. Hence, there are already safeguards in place for some commercial services. These safeguards seem to be less sensitive for impersonation of age and gender. However, they prevent us from reliably evaluating the underlying LLM.

# E    Limitations

Our vision based experiments are a two step process. Thus, a limitation of our work is that the results on the vision datasets fundamentally depend on the performance and biases of the VLM models as well. We try to alleviate this fact by evaluating multiple different CLIP variants. Additionally, the results obtained with proprietary models such as ChatGPT may be hard or costly to reproduce and the training regime and data as well as the systems prompts are unknown.