# OpenReview forum: "In-Context Impersonation Reveals Large Language Models' Strengths and Biases"
_NeurIPS.cc/2023/Conference — NeurIPS 2023 spotlight_

### Official Review · Reviewer_ZdXm · 2023-06-28

**Soundness:** 3 good
**Presentation:** 3 good
**Contribution:** 2 fair
**Rating:** 7
**Confidence:** 3

**Summary:**

This paper explores the ability of a LLM to engage in roleplay, and how being prompted to do so influences the models' generations. In particular, they focus on exploring how models prompted to behave like children seem to do so accurately using comparisons from the human developmental psychology literature.

**Strengths:**

**Originality:** This paper is not particularly original, but that is not something I count against it. Well executed experimental evaluations of "common knowledge" are an essential part of science.

**Clarity:** The paper is clearly and compellingly written.

**Significance:** It is difficult to assess the significance of this paper due to the concerns about the experimental results I identify below. If the authors' claims are correct then the paper would be highly significant.

**Weaknesses:**

**Over-Claiming**

The paper makes several false or overstated claims about its results and those of others. For example, it characterizes its relationship with the existing psychology literature as follows:

> Importantly, LLMs impersonating older participants generate higher average rewards (β = 0.15, p < .001), thereby replicating a general pattern found in the developmental literature [59].
>
>  Lastly, we analyze how regression weights of the probit-regression were influenced by the age group the LLM is impersonating. Figure 2 (bottom right) reveals that LLMs pretending to be older explored their environment less (β = −0.06, p < .001) and exploited more (β = 0.05, p < .001). This pattern is in line with several results from the psychological literature which also found that children explore their environment more [60]. These results suggest that impersonating LLMs can recover human-like developmental stages of exploration.

These results are not obviously consistent with the claims made in [59] and [60]. For example, [60] finds "... the two groups of children did not differ significantly in their extent of generalization" and "... exploration did not reliably differ by age group." The only age group studied in this paper that would be considered an adult by [60] is the 20-year olds, and its not at all clear that the effects in this paper are qualitatively or quantitatively similar to the ones in [60]. For [59], I am unable to identify which claims specifically the authors think that they reproduce. These problems are exacerbated by the fact that the authors do not release the raw data for me to analyze.

The authors claim "Vicuna is competitive with proprietary services such as ChatGPT," however according to their own source users prefer the results from ChatGPT to Vicuna over 60% of the time. Additionally, when looking at how likely a user is to find the two models substitutable, there is consistently a gap of around 10% in how often users prefer ChatGPT over Model M compared to how often they prefer Vicuna over Model M for each third Model M in the benchmark.

The authors claim

> For both LLMs, in both datasets, we observe that with increasing age, the complexity of the vocabulary and the attributes of the mentioned objects increases. A 2-year-old persona talks about the sound the bird or the car makes, the shapes of the wings or wheels, and the emotions attached to seeing or riding it. A 4-year-old persona interestingly mentions experiences seeing the bird or the car more distinctly. A 7-year-old persona starts using more complicated adjective phrases, e.g. can drive on rough roads and outside places, whereas a 13-year-old persona takes it one step further, e.g. brownish-gray body with distinctive rusty colored markings. Finally, a 20-year-old persona makes a more complete description of the object including where the bird is found or what the car is mainly used for. This is in line with [68] where the authors show that given the same length of text, smaller children use less diverse vocabulary, repeat a lot, and focus on non-academic vocabulary.

**Missing Citations**

[Prompt Programming for Large Language Models: Beyond the Few-Shot Paradigm](https://arxiv.org/abs/2102.07350) was one of the earliest papers exploring personas in LLMs. It introduces a "Master Translator" prompt and finds that performance on translation tasks improves when using it. The same authors also wrote [Role-Play with Large Language Models](https://arxiv.org/abs/2305.16367), which is likely simultaneous with this paper and not something I judge the authors negative for omitting. However I think that the paper would be enhanced by drawing on some of the content in the paper.

The authors touch on Text-to-Image and VLMs, but largely omit reference to the extensive literature on using persona-like prompts to steer image generation. Most relevant is [VQGAN-CLIP](https://arxiv.org/abs/2204.08583) which explicitly discusses this phenomenon in terms very similar to the submitted paper, including prompting the model to generate "a child's drawing of an X." Other papers such as [GLIDE](https://arxiv.org/abs/2112.10741) and [ImaGen](https://arxiv.org/abs/2205.11487) also prompt the models to impersonate specific artists and/or their styles.

**Questions:**

1. How and why were the particular experimental settings used chosen? For example, you ask the model to impersonate a 2, 4, 7, 13, 20-year-old. Why those numbers and not others? Why does the temperature change between single-token and multi-token generations?

2. Your prompt for MMLU ("Please consider the following multiple-choice question and the four answer options A, B, C, and D. Question: {task}") differs from the official implementation of MMLU. Why was this chosen? **Can you report the result you using the official prompt?((

3. You write "We focus on 32 tasks (19 STEM and 13 Humanities) and for each task we compute the task accuracy averaged over all task questions (95% confidence intervals are computed over the average task accuracy). We compare the task expert results with the average of all domain expert personas, the average of all non-domain expert personas and the random baseline (horizontal line)." Why were these 32 tasks out of the MMLU benchmark chosen? Why did you not use all of the tasks? **Can you report the results on all tasks**

4. You write

> We also observe that impersonation can reveal biases encoded in the LLMs. A race bias becomes apparent when we ask the LLMs to impersonate a “black” or “white” person. ChatGPT tends to describe both birds and cars better when posing as a white person. Vicuna-13B, on the other hand, provides better descriptions of cars as a black person. Gender biases are a bit less noticeable, but we still find Vicuna-13B giving better bird descriptions as a woman persona and ChatGPT identifying cars better as a man persona. While instruction-based fine-tuning [54] tries to remedy social biases encoded in LLMs to some extent, we can still expose them through in-context impersonation.

This exposes a *difference* but it's unclear if it manifests a social bias. **Can you explain how these results connected to social biases?**

5. What do you believe the primary contribution(s) of the paper to be?

**Limitations:**

The authors did adequately.

---

> ### Author Rebuttal · Authors · 2023-08-09
>
> We thank the reviewer for their feedback on the point regarding association to child development. However, we disagree with the fact that our results are inconsistent with those papers. Our results are similar to the idea described in these papers that children tend to explore their environments in a more directed fashion than adults. It is, by now, a well-established finding that children tend to show a higher tendency to engage in directed exploration than adults.
>
> [60] studied both exploration and generalization across different age groups. They found that both directed and random exploration were reliably higher for children compared to adults. Similarly, we find increased directed exploration, as measured by the effect of relative uncertainty, for LLMs impersonating younger ages. [59] has a section on “directed exploration” which corresponds to the effect of relative uncertainty in our probit regression. This whole section is about how children tend to use directed exploration more than adults.
>
> To further link our results to the literature on developmental psychology, we have modified this section: _This pattern is in line with several results from the psychological literature which also found that children show higher levels of directed exploration than adults [60]. These results suggest that impersonating LLMs can recover human-like developmental stages of exploration in a two-armed bandit task. If life is seen as an exploration-exploitation problem, then younger agents should show higher amounts of directed exploration [F,G]._
>
> Lastly, we ensure reproducibility (including data generation) by releasing our code which is already included in the supplementary.
> [F] _Developmental changes in learning resemble stochastic optimization_, Giron, A. P., Ciranka, S., Schulz, E., van den Bos, W., Ruggeri, A., Meder, B., & Wu, C. M. (accepted). Nature Human Behaviour.
>
> [G] _Systematic exploration and uncertainty dominate young children's choices_, Blanco, N. J., & Sloutsky, V. M. (2021). Developmental Science, 24(2), e13026.
> ### Vicuna vs ChatGPT
> We will adjust that Vicuna is competitive with ChatGPT in some domains. Evaluating one of the strongest open source models at the time of submission is beneficial: results are easier to reproduce, which is important overall. Our observations on the trend are very similar for both models.
>
> ### LLM vocabulary & grammar
> Rather than arguing that LLMs faithfully represent the language of children, we qualitatively observe interesting patterns similar to humans. In our supp Sec3.1 with several text complexity metrics we quantitatively showed that the LLM descriptions of older personas are more complex than younger personas.
>
> ### Missing Citations
> The conceptual “Role-Play with Large Language Models” paper (no experiments presented) was not publicly available at the time of our submission. It argues that some of LLMs’ behaviors could be better understood as role-taking. We on the other hand, across the cognitive, language, and vision domains extensively evaluate the behavior of different LLMs when they were explicitly asked to take on different personas.
> We adopted the meta-prompting strategy suggested in “Prompt Programming for Large Language Models: Beyond the Few-Shot Paradigm” to create variations of our impersonation prompts to study robustness w.r.t. impersonation prompt variations (results in the prompt variations section in the general comment).
> We will add these two papers in our related work.
>
> We have not investigated text-to-image models (there are many) as it is out of our scope but will mention them in our related work section, as they are also prompted with text.
>
> ### Questions
>
> 1. *Age groups & temperature:* We chose key developmental stages of early childhood, childhood, adolescence, and adulthood. Nonetheless, we reran our experiments (with all prompt variations) for more ages. We discuss these results in the general comment, where we find a significant effect of age on the reward for ages until 20 (β=0.15,p<.001), but no significant effect from ages 20-60 (β=-0.01,p<.1).
> Different temperatures were chosen based on the recommended default values of each LLM as this will reflect the most common use case but the same value may not generalize across LLMs, e.g. text properties depend on a variety of factors including training data, model scale, model architecture.
> 2. *Official MMLU prompt:* Several MMLU evaluations [57, H], lead to small variations when comparing different models’ ranks. We include results with the MMLU official prompt (see Figure 1 under Orig. Ctx 1 IP), i.e. by using the MMLU prompt at the start and keeping our impersonation strategy. Our expertise-based impersonation trends still hold, and the absolute accuracy values improve. Indeed, our findings on impersonation are not dependent on the formulation of the task.
> 3. *All MMLU tasks:* Due to space restrictions, we focus on the STEM and Humanities tasks but the expertise trends hold for the other domains. We include results for all domains in Figure 3, e.g for STEM the non-domain experts are all the non-STEM experts.
> 4. *Social Biases:* We agree that the connection to stereotypical social biases is ambiguous. Our goal is not to test for alignment between LLM biases and human social biases, but to show the existence of LLM biases in sensitive categories such as gender and race. We will adjust this statement.
> 5. *Primary contributions:* We systematically investigate impersonation over multiple domains (cognitive, visual, language) and architectures (VLMs + LLMs) which is further strengthened by the findings in this rebuttal. We show that dataset specific expertise can improve performance, but biases also exist.
>
> [H] _Holistic Evaluation of Language Models_, Bommasani, Rishi, Percy Liang, and Tony Lee. Annals of the New York Academy of Sciences (2023).

---

> > ### Comment · Reviewer_ZdXm · 2023-08-15
> > **Excellent rebuttal, I now vote for acceptence**
> >
> > > [60] studied both exploration and generalization across different age groups. They found that both directed and random exploration were reliably higher for children compared to adults. Similarly, we find increased directed exploration, as measured by the effect of relative uncertainty, for LLMs impersonating younger ages. [59] has a section on “directed exploration” which corresponds to the effect of relative uncertainty in our probit regression. This whole section is about how children tend to use directed exploration more than adults.
> > >
> > > To further link our results to the literature on developmental psychology, we have modified this section: This pattern is in line with several results from the psychological literature which also found that children show higher levels of directed exploration than adults [60]. These results suggest that impersonating LLMs can recover human-like developmental stages of exploration in a two-armed bandit task. If life is seen as an exploration-exploitation problem, then younger agents should show higher amounts of directed exploration [F,G].
> > >
> > > Lastly, we ensure reproducibility (including data generation) by releasing our code which is already included in the supplementary.
> >
> > Thank you for the extended experiments, especially involving older individuals and multiple prompts. I find this substantially more compelling, and it's the primary reason for my change of opinion.
> >
> > > Vicuna vs ChatGPT
> >
> > I have zero objection to using Vicuna (though it's not "open source," it's publicly available under the LLaMA 2 license. My sole concern is that I think you're overselling its performance.
> >
> > > Rather than arguing that LLMs faithfully represent the language of children, we qualitatively observe interesting patterns similar to humans. In our supp Sec3.1 with several text complexity metrics we quantitatively showed that the LLM descriptions of older personas are more complex than younger personas.
> >
> > Prior to your new experiments, I did not believe that the results you reported were qualitatively similar to those obtained by the cited literature. Your new, extended experiments have largely ameliorated this concern.
> >
> > > The conceptual “Role-Play with Large Language Models” paper (no experiments presented) was not publicly available at the time of our submission.
> >
> > My apologies, I had misread the date on that paper. I do think it's worth mentioning but will not dock you for choosing to not.
> >
> > > We have not investigated text-to-image models (there are many) as it is out of our scope but will mention them in our related work section, as they are also prompted with text.
> >
> > > Age groups & temperature: We chose key developmental stages of early childhood, childhood, adolescence, and adulthood. Nonetheless, we reran our experiments (with all prompt variations) for more ages. We discuss these results in the general comment, where we find a significant effect of age on the reward for ages until 20 (β=0.15,p<.001), but no significant effect from ages 20-60 (β=-0.01,p<.1). Different temperatures were chosen based on the recommended default values of each LLM as this will reflect the most common use case but the same value may not generalize across LLMs, e.g. text properties depend on a variety of factors including training data, model scale, model architecture.
> >
> > These experiments were extremely compelling.
> >
> > > Official MMLU prompt
> >
> > The official MMLU prompt is `The following are multiple choice questions (with answers) about {TOPIC}.\n\n`, as seen [here](https://github.com/hendrycks/test/blob/master/evaluate.py).
> >
> > > MMLU tasks: Due to space restrictions, we focus on the STEM and Humanities tasks but the expertise trends hold for the other domains. We include results for all domains in Figure 3, e.g for STEM the non-domain experts are all the non-STEM experts.
> >
> > The standard thing to do (see, e.g., GPT-NeoX-20B, LLaMA, LLaMA 2, and PaLM) is to report accuracies averaged over each of the four domains. That's what is recommended by the MMLU paper, and would give you plenty of space.
> >
> > > Social Biases
> >
> > Perhaps "differences" would be a more appropriate word? I do think that it's important to not call them social biases in the absence of evidence of power.
> >
> > **New score:** 7

---

> > > ### Author Response · Authors · 2023-08-16
> > >
> > > We would like to sincerely thank the reviewer for their timely response and for raising their score to 7 (Accept). We are glad to hear that our response has clarified their concerns. Below we describe how we will improve our manuscript further based on their feedback.
> > >
> > > #### Vicuna vs ChatGPT
> > >
> > > We will update our submission to better reflect Vicuna’s capabilities as promised in the original rebuttal comment.
> > >
> > > #### MMLU
> > >
> > > The results in our rebuttal with the official MMLU prompt (c.f. rebuttal pdf Figure 1 left “Orig. Ctx”) are using this exact prompt.
> > >
> > > We agree that it is important to show all four domains and will update the main paper with the results from the rebuttal accordingly.
> > >
> > > #### Social Biases
> > >
> > > We will avoid the term “social bias” and replace it with the term “differences” where appropriate, i.e., when talking about our results and findings. In our opinion, the term “bias” by itself is still a fitting term for some of the effects we observe from in-context impersonation. However, we will make sure our definition of the term is clear (i.e. data-induced differences) and that it is not to be confused with stereotypical social biases.
> > >
> > > We hope this clarifies the remaining open questions of the reviewer and we would be happy to discuss further if there are any other open questions from Reviewer ZdXm or any other reviewers.

---

### Official Review · Reviewer_KqkQ · 2023-07-02

**Soundness:** 3 good
**Presentation:** 3 good
**Contribution:** 3 good
**Rating:** 7
**Confidence:** 4

**Summary:**

This paper tries to understand whether LLMs can take on different personas when they do in-context generation. They have tested different settings: In a two-armed bandit setting, they ask LLMs to act as different-aged people; in a reasoning task, LLMs are asked to behave as different-level experts; and in a vision-and-language task, multiple roles have been set to the LLMs. The authors find while impersonation can improve model performance, such methods also reveal the biases in LLMs.

**Strengths:**

- Test their hypothesis across different domains and with various persona aspects.
- They have set the persona specific to each task, e.g, expertise-based persona for reasoning ability analysis.
- Well-presented results.

**Weaknesses:**

- LLMs seem to be sensitive to prompts [1]. And I'm wondering how stable the conclusion will be.
- I'm curious to what extent the LLMs align with human behavior. For example, in the two-armed bandit task, the authors observe "LLMs could reproduce human-like developmental stages of exploration behavior". And I'd be curious how humans (from different ages) perform the same task. Similar to other tasks.

[1] Arora, Simran, et al. "Ask me anything: A simple strategy for prompting language models." arXiv preprint arXiv:2210.02441 (2022).

**Questions:**

- What if in each setting, you add a random persona? For example, in the domain expert case, you impersonate LLMs as "normal student" (or as "specialist"). I'm just curious how certain tokens ("expert") will impact the result.

---

> ### Author Rebuttal · Authors · 2023-08-09
>
> We thank the reviewer for their insightful comments indicating that we tested our hypothesis across several different domains and with various persona aspects and presented the results well.
>
> ### Prompt sensitivity
> Please see the general comment for a detailed analysis of prompt sensitivity where we show that our findings are robust to prompt variations across the three tasks we present.
> In addition to our findings that the impersonation results are robust to prompt variations, already showing a single failure case (e.g. LLMs have biases wrt to gender and race) may indicate a problem that should be addressed. Thus our conclusion that LLMs expose biases, would still hold.
>
> ### Human LLM alignment
> It is a fairly well-established observation that children tend to explore more in reinforcement learning tasks than adults. Past work such as the ones cited below mostly focused on comparing adolescent with adult behavior using three features for the probit regression, with one additional feature quantifying the amount of random exploration. In our work, we were interested in how exploration differs between LLMs impersonating children and LLMs impersonating adults. Moreover, our own simulations assessing how recoverable the original three features are showed that one can really only reliably measure the amount of exploitation and directed exploration from the regression, which is what we ended up doing. Our findings of increased directed exploration in children align well with the literature on human exploration, for example, as reported in:
>
> _Children are more exploratory and learn more than adults in an approach-avoid task_, Liquin, E. G., & Gopnik, A. (2022). Cognition, 218, 104940.
>
> _Searching for rewards like a child means less generalization and more directed exploration_, Schulz, E., Wu, C. M., Ruggeri, A., & Meder, B. (2019). Psychological science, 30(11), 1561-1572.
>
> _Systematic exploration and uncertainty dominate young children's choices_, Blanco, N. J., & Sloutsky, V. M. (2021). Developmental Science, 24(2), e13026.
>
> ### Random persona
> We include a discussion regarding what we term the neutral baseline (to distinguish from the random baseline, e.g. 25% on MMLU) in the general comment. This baseline considers the average between four personas (student, average student, person, and average person) for MMLU and two personas for the vision tasks (person and average person). We show that, as expected, the neutral baseline performs worse than the respective expert.

---

> > ### Comment · Area_Chair_kwbW · 2023-08-18
> >
> > Dear authors, thank you for submitting your responses to the reviewers comments.
> >
> > Dear reviewer KqkQ, after reading the authors' rebuttal do you have any additional thoughts?

---

> > ### Comment · Reviewer_KqkQ · 2023-08-20
> >
> > I have read the authors' rebuttal as well as others' reviews. Happy to increase the score to 7.

---

### Official Review · Reviewer_NysF · 2023-07-05

**Soundness:** 3 good
**Presentation:** 2 fair
**Contribution:** 2 fair
**Rating:** 6
**Confidence:** 4

**Summary:**

Language models can play different roles using prompting techniques (e.g., adding "you are a 3-year-old boy" at the end of the prompt). This paper investigates whether roles in the prompt affect language model performances on three tasks: multi-armed bandit, language-based reasoning, and visual category description. The paper found that:
1. Language models exploit more in multi-armed bandit tasks when prompted to be older.
2. Language models perform better when prompted to be task experts.
3. Language models describe objects better when prompted to be older. For certain object categories, language models show biases (e.g., describe cars better when prompted to be a man instead of a woman).

**Strengths:**

- The paper focuses on an interesting and novel topic: how role-playing (impersonation) affects language model performances on downstream tasks. This is a less-known research question in the NLP community.
- The finding about performance differences when describing certain object categories, such as cars, when prompting language models to be different genders provides a novel perspective to understand biases in language models and our society.
- The paper is generally well-written and can be easily understood. However, the writing should be more focused. For example, in the first paragraph of the introduction, I did not get any connection between Min et al. found prompting LLM with random labels will not degrade model performance and your research topic impersonation. Similarly, there is no need to introduce CLIP training when introducing the visual category description task.

**Weaknesses:**

- While the topic is to study how different roles affect language model performances, from experimental results, ages have the most significant impact on model performances, and other factors have much more minor effects on model performances. This seems obvious- it is easy to understand older people can exploit more in multi-armed bandit tasks and describe object categories more clearly. However, the most exciting finding, where gender/race can affect model performance on tasks, is not well studied. They only study two object categories from the visual object description task, and the performance difference between genders/races is much smaller and insignificant based on the plots. More systematic studies on language model biases during impersonation should significantly improve the work.

- Experimental results can become more solid using more prompt variations and model families. Language models are well-known to be sensitive to prompts; the authors should mention how robust their results are when changing some words in the prompt. Moreover, more model families besides Vicuna are appreciated, especially on bias experiments. Will the finding about gender bias hold for different model families?

- Besides the tasks with deterministic labels, it would also be interesting to show how impersonation affects task performances on non-deterministic tasks, such as social opinions or morality tasks. Since the major goal of impersonation is to simulate different people, it would be more interesting and convincing to show that language models indeed reflect these group opinions on non-deterministic tasks when prompted to be them.

**Questions:**

- What are the dashed lines in Figure 3 (results on STEM and Humanities)?

**Limitations:**

See weaknesses. I'm happy to increase my rating if these are well addressed.

---

> ### Author Rebuttal · Authors · 2023-08-09
>
> We thank the reviewer for their insightful comments indicating that we focus on an interesting and novel topic, that is a less-known research question in the NLP community and provides a novel perspective to understand biases in language models and society.
> ### Triviality of the age effect
> While it seems obvious that older humans perform better in these tasks (i.e. they exploit more in multi-armed bandit tasks), to the best of our knowledge we are the first to show that LLMs replicate similar trends when using in-context impersonation. Even though this finding might seem obvious in hindsight, it is not intuitive why this would also happen in LLMs.
> ### Significance of the results
> We run $\text{Chi}^2$ tests for the race and gender results presented in Figure 5 of the rebuttal PDF. Thus, we consider 3 CLIP models, 5 different seeds and 6 different impersonation prompt variations. We find that for all experiments considered, {CUB, Stanford Cars} x {man/woman, black/white}, p<0.001. Thus, we conclude that our results are significant.
> ### Biases for more object categories
> To clarify, CUB dataset contains 200 different bird categories (11K images) and Stanford Cars contains 196 categories of cars (16K images) with various sizes, appearances, habits etc and the visual classification task in these datasets (due to the large number of categories and fine-grained similarities between these objects) is far from solved. While a sentence on distinguishing properties of a bottle and a car can be very short (these objects have many uncommon properties), a sentence describing the differences between a black-footed albatross and laysan albatross that also remains distinguishing between a field sparrow needs to talk about many more category-specific properties. Arguably, such a fine-grained categorization task is one of the most challenging ones even if we work only in the language domain. Nevertheless, we extend our analysis to other datasets and more categories by using FGVC Aircraft [B] (100 categories of aircrafts from different manufacturers and eras) and Oxford flowers [C] (102 categories of flowers with large scale, pose and light variations).
> For gender, we find significant performance differences when evaluating the descriptions generated by Vicuna-13B on the two additional datasets, strengthening our original argument that these LLMs exhibit biases (Figure 4). This means that descriptions generated by female personas outperform those generated by male personas across all three tested VLMs. For racial biases, we see only smaller differences across evaluation with different VLMs. The same trends hold for ChatGPT.
> ### Prompt sensitivity
> Please see the general comment for a detailed analysis of prompt sensitivity where we show that our findings are robust across all three tasks (bandit, MMLU, and vision descriptions).
> ### Do findings transfer to other LLM families?
> Most open-source models such as Vicuna are fine-tuned from the same base model. Recently, a new foundational open-source model, Llama 2 (70B, Chat variant) has been released which is significantly larger than Vicuna-13B and has been trained on more data. We rerun the bandit experiments using Llama 2 and come to the same conclusions. The effect of age in the range of 2-20 on the reward is β=0.16 (p<.001) and β=0.26 (p<.001) for Vicuna-13B and Llama 2, respectively.
>
> Additionally, we ran our experiments using the proprietary Google PaLM model via their API. When we apply in-context impersonation of racial biases, the API does not provide an answer because the LLM model output is flagged by a text classifier to be unsafe. Hence, there are already safeguards in place for some commercial services.These safeguards seem to be less sensitive for impersonation of age and gender. However, they prevent us from reliably evaluating the underlying LLM. We will add Llama 2 results and as well as a brief discussion on these existing safeguards to the paper.
> ### Non-deterministic tasks
> Related work has explored non-deterministic tasks, e.g. [D] finds that LLMs give inhumanly accurate answers specifically in open-ended questions whereas a collection of human answers is much more diverse. [E] studies group opinions via LLM-generated survey responses and found that those “currently lack the signals found in human populations”. Additionally, we discussed [40] in our related work section (lines 78-79), which simulates the survey responses of sub-populations. In this work we focus on diverse tasks that can be objectively measured. Hence, we believe expanding the set of experiments to non-deterministic tasks is out-of-scope for this work, but see them as a relevant future direction.
> ### Dashed lines in Figure 3
> The dashed lines in Figure 3 correspond to the random baseline, i.e. 25% for a 4 choice exam question. We will update the caption accordingly to clarify this.
>
> #### References
> [B]: _Fine-Grained Visual Classification of Aircraft_, S. Maji, J. Kannala, E. Rahtu, M. Blaschko, A. Vedaldi, arXiv.org, 2013
>
> [C]: _Automated Flower Classification over a Large Number of Classes_, M.-E. Nilsback, and Andrew Zisserman. 2008 Sixth Indian Conference on Computer Vision, Graphics & Image Processing (2008): 722-729.
>
> [D]: _Using Large Language Models to Simulate Multiple Humans and Replicate Human Subject Studies_, G. Aher, R. I. Arriaga, A. T. Kalai, ICML 2023
>
> [E]: _Questioning the Survey Responses of Large Language Models_, R. Dominguez-Olmedo, M. Hardt, C. Mendler-Dünner, arXiv, 2023

---

> > ### Comment · Area_Chair_kwbW · 2023-08-18
> >
> > Dear authors, thank you for taking the time to respond to the comments.
> >
> > Dear reviewer NysF, did the authors adequately address your concerns?

---

> > > ### Comment · Reviewer_NysF · 2023-08-18
> > > **Thanks for the rebuttal**
> > >
> > > I thank the authors for providing such a detailed response. The new experiments on different prompts / model families and the new significance test are very convincing. Therefore, I increase my score to 6 regarding this.

---

> > > > ### Author Response · Authors · 2023-08-19
> > > >
> > > > We would like to thank the reviewer for their response and increasing their rating to 6 (Weak Accept). We are happy that our additional results on prompt variations, different model families and the provided significance tests are very convincing. We will include these results in the final manuscript.

---

### Official Review · Reviewer_J5Gp · 2023-07-06

**Soundness:** 3 good
**Presentation:** 3 good
**Contribution:** 3 good
**Rating:** 6
**Confidence:** 3

**Summary:**

This paper presents a comprehensive evaluation on the downstream impact of impersonation with the use of LLM and multimodal model (specifically CLIP). There is a couple of interesting finding that LLM, when impersonated, aligns to some behavior carried by the people in the respective norm. This happens in ages, races, genders, and degree of expertises. The finding seems to suggest that one can exploit such phenomenon to make a large model perform better on some downstream tasks. This paper also suggest that such exploitation should be concerned. The arguments are of conflict, but also interesting.

**Strengths:**

This paper introduced three types of evaluation using recent LLM and multimodal CLIP to probe LLM's downstream performances: exploration-exploitation, reasoning (MMLU), and shallow multimodal understanding. There is an very interesting finding that impersonation aligns to different behavior and downstream performances on these benchmarks. An immediate takeaway is that it gives an "interface" to easily boost downstream performance with a discovered stereotype to impersonate. The respective concern raised in this paper is also of value to the community. In general, I am happy to see such argument and findings to appear in the conference, as long as the argument goes well.

**Weaknesses:**

A few downsides:
- On one hand, this paper seems to suggest one can easy explore stereotypes to impersonate and use them to improve some downstream performances. On the other hand, this strategy is of special concern, and I believe, should be avoided. ***It's unclear what this paper's take is.*** A very simple and immediate counter argument to this paper is that, such exploitation aggregates the existing stereotypical biases in the real world. And if this concern is valid, then this paper shouldn't advertise it as a strength.

- Not clear if the observation is robust against prompt variance. The only place seems to have robust eval is the two-armed bandit experiment.

- The age-related evaluation is narrow. A more comprehensive range should go beyond age 20. This can be useful to probe for potential age biases and have a more comprehensive picture about the age "mimicking" phenomenon.

**Questions:**

Please see the general comments.

**Limitations:**

Implicitly or explicitly, this paper suggests the exploitation of stereotypes to improve model performances on downstream tasks.

---

> ### Author Rebuttal · Authors · 2023-08-09
>
> We would like to thank the reviewer for their helpful comments indicating that they would be happy to see our findings appear in the conference, and we will use the rebuttal to further strengthen our argument.
>
> ### Broader Impact
> By “strength” we refer to the improved performance when impersonating experts. However, we agree with the reviewer’s take that these still fall under biases that ideally should not be present in LLMs.
> We specifically discourage crafting (system) prompts for maximal performance by exploiting biases, as this may have unexpected side effects, reinforce societal biases and poison training data obtained with such prompts. Aside from the directions mentioned in our conclusion (lines 348-349; e.g. how do these biases emerge as a function of scale), we believe that our submission will make LLM providers aware that they should quantitatively test for these biases. We will update our discussion to clarify our stance and will adjust our narrative not to highlight any of these biases as strengths.
>
> ### Prompt variance
> Please see the general comment for a detailed analysis of prompt variance where we show that our findings are robust not only in the bandit task, but also for the reasoning (MMLU) and visual description experiments across a large set of automatically generated prompt variations.
>
> ### Age evaluation
> We address the concerns regarding age evaluation in the general comment, where we show that the effect of age replicates if all ages between 2-20 are evaluated and diminishes in the 20-60 range for the bandit task.

---

> > ### Comment · Reviewer_J5Gp · 2023-08-18
> > **Author's take and argument on the exploitation of biases should be clearly and non-briefly addressed in the draft**
> >
> > Thank you for the rebuttal. I agree that the exploitation of biases can be a measurement for LLM providers and this actually can be a good use case. However, this point should be clearly addressed in the draft. Emphasis on downstream performance improvement is essentially an aggregation of biases instead of an effort of mitigation.

---

> > > ### Author Response · Authors · 2023-08-19
> > >
> > > We would like to sincerely thank the reviewer for their response.
> > >
> > > To clearly address the impact of our work, we will add a dedicated _Broader Impact_ section in our draft, that includes our impact discussion from the rebuttal.

---

### Author Rebuttal · Authors · 2023-08-09

# General Response

We thank the reviewers for their helpful and insightful comments. As summarized by reviewer J5Gp, NysF and KqkQ, we present a comprehensive evaluation on the downstream impact of impersonation/roleplay with the use of LLM (two variants) and multimodal model (three variants of clip) in three tasks, namely multi-armed bandit (and show exploration-exploitation capabilities), language-based reasoning (and show expertise capabilities), and visual category description (and show multimodal understanding capabilities). To the best of our knowledge, ours is the first systematic evaluation that shows LLM, when impersonated, aligns to some behavior carried by the people (as pointed out by reviewer J5Gp) in the respective norm.

All reviewers (ZdXm,NysF,KqkQ,J5Gp) agree that our submission is clearly and compellingly written. Focusing on an interesting and novel topic that is less-known to the NLP community (NysF), our well executed experimental evaluations (ZdXm) provide a novel perspective to understand biases in language models and our society (NysF). Our approach to test our hypothesis across different domains and with various persona aspects by setting the persona specific to each task was well presented (KqkQ). A very interesting finding is that impersonation aligns to different behavior and downstream performances on three benchmarks (J5Gp). Reviewer ZdXm said “if the authors' claims are correct then the paper would be highly significant”. With this rebuttal, we would like to show with various additional experiments and extended discussions that our claims are indeed correct.

Our additional experiments requested by the reviewers helped us to clearly show that,
1. Our framework is robust to automatic prompt variations for in-context impersonation. Our results are independent of how the MMLU task is phrased and presented to the LLMs.
2. Our observations hold for the age effect when evaluating different specific ages and for the expertise effect when evaluating on additional vision datasets and MMLU tasks.

### Robustness to prompt variations (common request):
We show that our findings are robust to prompt variation of in-context impersonation and will update our manuscript accordingly. We follow the meta-prompting approach from [A] as suggested by reviewer ZdXm and automatically created 5 prompt variations:
- Should you be transformed into a {persona}
- Imagine you are a {persona}
- Should you assume the role of a {persona}
- Were you to take on the persona of a {persona}
- In the case of you being a {persona}

We ran all our experiments (multi-armed bandit, reasoning and vision tasks) with these additional prompts and averaged over the 6 prompt variations.

In the bandit task, we see a significant effect of age on the received reward which was β=0.15 (p<.001) for “If you were a {persona}” and β=0.16 (p<.001) with all prompt variations. Similarly, the same effect of age on exploration (β=−0.06, p<.001) and exploitation (β=0.05, p<.001) can still be observed on average for all prompts: β=−0.03, p<.001 and β=0.04, p<.001, respectively.

On the MMLU reasoning task (see Fig. 1 _Our Ctx 6 IP_), the expertise trend remains the same with these additional prompt variants, i.e. the task expert outperforms the domain expert which, in turn, outperforms the non-domain expert.

In the vision tasks, for CUB we see the same significant effect of age (see Fig. 5 top) that we had originally reported: Descriptions generated by older personas outperform those by younger personas. Albeit decreased in effect size, we find that expertise and race has an effect on downstream performance. Conversely, the effect of gender is even larger than what we had previously reported.
On Stanford Cars (Fig. 5 bottom) we see slightly reduced effects of race and gender, but the effect of expertise is very clear. The age effect, that older personas produce better descriptions, becomes even more clear.

### Neutral baseline (requested by reviewers KqkQ and DtKQ (ethics))
We include results on the reasoning task with the neutral baseline in Fig. 1. We consider the average of 4 neutral personas: student, average student, person, and average person. As expected, the neutral persona performs worse than or on par with the non-domain expert. We will update the supplementary with results for all domains, all of which show the same behavior.
We also evaluate appropriate neutral personas for the vision task, namely average person and person. On CUB we find for Vicuna-13B that the dataset specific expert outperforms the average neutral persona.

### Age evaluation (requested by reviewers ZdXm and J5Gp)
We ran additional experiments with Vicuna-13B (all prompt variations) extending the age-range beyond age 20 up to age 60 on the bandit task. We consider steps of 2 from 2 until 30 and steps of 5 from 30 until 60. The effect of age on the task performance diminishes after the age of 20 and becomes more stable. While between ages 2-20, there is a clear effect of age on the reward (β=0.16,p<.001), between ages 20-60, the effect is not significant (β=-0.01,p<.1).
From age 2 to 20 exploration decreases (β=0.04,p<.001) while exploitation increases (β=0.03,p<.001). After the age of 20, both exploration and exploitation decrease (β=-0.02,p<.001 and β=-0.02,p<.001).
We also evaluate more ages on the vision task and observe a similar trend. On CUB, we observe a positive effect on the classification accuracy for ages 2 to 20 (β=0.69,p<.001), but the effect is insignificant for ages 20 to 60 (p-value of 0.101).
In conclusion, these observations further strengthen our initial claims that LLMs are able to impersonate the behavior of early developmental stages of humans. We will add these results and a discussion to the paper.

[A]: _Prompt Programming for Large Language Models: Beyond the Few-Shot Paradigm_, Reynolds, L. and McDonell, K. Extended Abstracts of the 2021 CHI Conference on Human Factors in Computing Systems (2021).

---

### Decision · Program_Chairs · 2023-09-21

**Decision:**

Accept (spotlight)

**Comment:**

The authors explore the effects of roles (social identity, domain expertise) in LLM instructions on downstream tasks. The findings are interesting, showing that models respond differently when instructed to follow different roles for the same task.

All reviewers agree that this is quality work, pointing out the timeliness of this study, the impact it can have in our society, and the clarity of presentation. The reviewers, however, also raised some concerns regarding the robustness of the method, the role of age, as well as some ethical concerns. The authors wrote comprehensive responses to reviewers and ethical committee and most concerns were addressed. For these reasons, I am happy to accept this paper.